



# Biomass burning nitrogen dioxide emissions derived from space with TROPOMI: methodology and validation

Debora Griffin[1], Chris A. McLinden[1,2], Enrico Dammers[3], Cristen Adams[4], Chelsea E. Stockwell[5,6], Carsten Warneke[5], Ilann Bourgeois[5,6], Jeff Peischl[5,6], Thomas B. Ryerson[5,a], Kyle J. Zarzana[7], Jake P. Rowe[6,7], Rainer Volkamer[6,7,9], Christoph Knote[8,b], Natalie Kille[6,7,9,c], Theodore K. Koenig[6,7,d], Christopher F. Lee[7], Drew Rollins[5], Pamela S. Rickly[5,6], Jack Chen[1], Lukas Fehr[2], Adam Bourassa[2], Doug Degenstein[2], Katherine Hayden[1], Cristian Mihele[1], Sumi N. Wren[1], John Liggio[1], Ayodeji Akingunola[1], and Paul Makar[1]

[1]Air Quality Research Division, Environment and Climate Change Canada, Toronto, Ontario, Canada
[2]Department of Physics and Engineering Physics, University of Saskatchewan, Saskatoon, Saskatchewan, Canada
[3]Netherlands Organisation for Applied Scientific Research (TNO), Climate Air and Sustainability (CAS), Utrecht, The Netherlands
[4]Resource Stewardship Division, Government of Alberta, Alberta Environment and Parks, Edmonton, Alberta, Canada
[5]NOAA Earth System Research Laboratories (ESRL), Chemical Sciences Laboratory, Boulder, CO, USA
[6]Cooperative Institute for Research in Environmental Sciences, University of Colorado Boulder, Boulder, CO, USA
[7]Department of Chemistry, University of Colorado Boulder, Boulder, CO, USA
[8]Meteorological Institute, LMU Munich, Munich, Germany
[9]Department of Atmospheric and Oceanic Sciences, University of Colorado Boulder, Boulder, CO, USA
[a]Now at: Scientific Aviation, Boulder, CO, USA
[b]Now at: Model-based Environmental Exposure Science, Faculty of Medicine, University of Augsburg, Augsburg, Germany
[c]Now at: Institute of Energy and Climate Research: Troposphere (IEK-8), Forschungszentrum Jülich GmbH, 52425 Jülich, Germany
[d]Now at: College of Environmental Sciences and Engineering, Peking University, Beijing, 100871, China

**Correspondence:** D. Griffin (debora.griffin@canada.ca)

**Abstract.** Smoke from wildfires is a significant source of air pollution, which can adversely impact air quality and ecosystems downwind. With the recently increasing intensity and severity of wildfires, the threat to air quality is expected to increase. Satellite-derived biomass burning emissions can fill in gaps in the absence of aircraft or ground-based measurement campaigns, and can help improve the on-line calculation of biomass burning emissions as well as the biomass burning emissions inventories that feed air quality models. This study focuses on satellite-derived $NO_x$ emissions using the high-spatial resolution TROPOspheric Monitoring Instrument (TROPOMI) $NO_2$ dataset. Advancements and improvements to the satellite based determination of forest fire $NO_x$ emissions are discussed, including information on plume height and effects of aerosol scattering on the satellite-retrieved vertical column densities. Two common top-down emission estimation methods, (1) an Exponentially Modified Gaussian (EMG) and (2) a flux method, are applied to synthetic data to determine the accuracy and the sensitivity to different parameters, including wind fields, satellite sampling, noise, lifetime and plume spread. These tests show that emissions can be accurately estimated from single TROPOMI overpasses. The effect of smoke aerosols on TROPOMI $NO_2$ columns (via AMFs) is estimated and these satellite columns and emission estimates are compared to aircraft observations from four different aircraft campaigns measuring biomass burning plumes in 2018 and 2019 in North America. Our results



indicate that applying an explicit aerosol correction to the TROPOMI $NO_2$ columns improve the agreement with the aircraft observations (by about 10-25 %). The aircraft- and satellite-derived emissions are in good agreement within the uncertainties. Both top-down emissions methods work well, however, the EMG method seems to output more consistent results and has bet-ter agreement with the aircraft-derived emissions. Assuming a Gaussian plume shape for various biomass burning plumes, we

estimate an average $NO_x$ e-folding time of $2\pm1$ h from TROPOMI observations. Based on chemistry transport model simula-tions and aircraft observations, the net emissions of $NO_x$ are 1.3 to 1.5 times greater than the satellite-derived NO2 emissions. A correction factor of 1.3 to 1.5 should thus be used to infer net $NO_x$ emissions from the satellite retrievals of $NO_2$.

## 1    Introduction

Wildfires are a significant source of aerosols and trace gases in the global atmosphere (Andreae, 2019, and references therein).

Exposure to wildfire smoke has been associated with adverse health impacts and premature mortality (Matz et al., 2020). The health impacts are generally greater in close proximity to active fire areas, however, health impacts are also associated with long-range transport of smoke plumes (Matz et al., 2020). In recent years, the number of wildfires has increased (e.g. Romero-Lankao et al., 2014; Landis et al., 2018), primarily driven by droughts, higher temperatures, and fuel loading caused by tree death (e.g. Kitzberger et al., 2007; Littell et al., 2009; Westerling, 2016). Studies suggest the intensity of fires may continue to

rise driven by climate change and its associated droughts, higher temperatures, and an earlier spring season (Liu et al., 2013; Wotton et al., 2017). This increase in wildfires, combined with the focus on national emission targets and air quality monitor-ing, leads to an increasing demand for improved knowledge of wildfire emissions.

One type of pollutants emitted by wildfires is nitrogen oxides ($NO_x$ =$NO_2$+NO) which has adverse effects on the envi-

ronment and human health (Health Canada, 2018). $NO_x$ plays a significant role in the tropospheric production of ozone and can contribute to acid rain. Wildfire emissions of $NO_x$ exhibit large year-to-year variability and on average account for ap-proximately 15 % of the global $NO_x$ budget (Denman et al., 2007). The amount of nitrogen (N) released by wildfires strongly depends on the type of fuel being consumed (fuel nitrogen content) and the burning phase represented by the relative amounts of flaming and smoldering combustion. $NO_x$ is primarily emitted during flaming combustion at high temperatures, whereas

the release of reduced forms of nitrogen, such as $NH_3$, are favored during the lower-temperature smoldering phase (e.g. Goode et al., 2000; Burling et al., 2010; Roberts et al., 2020). Reactive nitrogen species are released through fuel pyrolysis, if the fire temperatures are below $\sim$1200°C (Roberts et al., 2020, and references therein); where radical chemistry within the flames con-verts these fuel N to oxidized nitrogen species and $N_2$ (Ren and Zhao, 2012; Roberts et al., 2020). Each wildfire is a mixture of different stages of combustion that can occur simultaneously or at various times and locations within a given wildfire perimeter

(Lindaas et al., 2021, and references therein).

Ground-based and aircraft measurements are difficult to obtain near the fire source (due to Temporary Flight Restriction zones) and field campaigns are infrequent with limited spatial coverage, while satellite-borne observations can be used to



constrain wildfire emissions and can provide emission estimates for fires missed by measurement campaigns. Satellite-remote sensing observations also have the advantage of near global coverage. Note that satellite-derived emission estimates (from satellite observations of pollutants and greenhouse gases) cannot be used alone to build total emission budgets for emission inventories, because they are limited by the presence of cloud cover, thick smoke, and the detection limit of the remote-sensing

instruments. Species that can be observed by satellite instruments and used to estimate fire emissions are also limited. Satellite-derived emissions can be derived using a variety of approaches, such as through the use of an inverse model or by directly using a mass balance or curve-fitting approach (de Foy et al., 2014). This study focuses on deriving the biomass burning emissions directly from satellite observations without the use of model simulations. Previously, de Foy et al. (2014) tested several different top-down emission estimation methods on synthetic data and concluded that emissions can be estimated accu-

rately within 5-40 %, across all methods. Global $NO_x$ emissions were first derived from satellite observations nearly 20 years ago by using a simple mass balance technique (Leue et al., 2001; Martin et al., 2003) applied to data from the Global Ozone Monitoring Experiment (GOME), 1995–2011, with a pixel size of $40 \times 320\,\mathrm{km}^2$ (Burrows et al., 1999). As satellites improved so did space-borne emission estimates, and in 2011 $NO_x$ emissions were derived for the first time on a city-wide scale (Beirle et al., 2011) using observations from the SCanning Imaging Absorption spectroMeter for Atmospheric CartograpHY (SCIA-

MACHY, 2002–2012, $30 \times 60\,\mathrm{km}^2$; Bovensmann et al., 1999). Further advances came with the Ozone Monitoring Instrument (OMI; 2004–present; $13 \times 24\,\mathrm{km}^2$; at nadir; Levelt et al., 2006; Krotkov et al., 2016), with which $NO_x$ emissions from cities, power plants and other point sources could be resolved (e.g., Beirle et al., 2011; Goldberg et al., 2019; Liu et al., 2020a). $NO_x$ emissions from large fires have also been derived from OMI observations (e.g., Mebust et al., 2011; Mebust and Cohen, 2014; Adams et al., 2019). More recently, Jin et al. (2021) reported TROPOMI-derived $NO_x$ emissions from biomass burning.

Good spatio-temporal coverage and high-spatial resolution enables a detailed plume shape, which is the key to accurately estimating fire emissions from satellite observations. With the recent advances in satellite-borne remote-sensing instruments, in terms of spatial resolution, as well as data product quality of the recorded spectra, top-down emission estimates can be improved. The Tropospheric Monitoring Instrument (TROPOMI), launched in October 2017, has a high enough spatial resolution

(3.5 km×5.5 km; 3.5 km×7 km prior to August 2019) that makes it possible to resolve single plumes (Griffin et al., 2019), and with this, satellite-borne remote-sensing observations have entered a new era.

The ultraviolet-visible (UV-vis) region, used to derive the nitrogen dioxide ($NO_2$) columns from TROPOMI observations, is influenced by aerosol scattering. This is a significant limitation when estimating fire emissions, since the TROPOMI obser-

vations near fires are almost always influenced by smoke aerosols. In most current operational retrieval algorithms for $NO_2$, an implicit aerosol correction is applied by assuming aerosols as effective clouds. This implicit aerosol correction is also applied for the operational TROPOMI air mass factor (AMF) (van Geffen et al., 2018). Previous studies showed that the implicit aerosol correction introduces a low bias of up to 50 % (e.g., Lin et al., 2014; Lorente et al., 2017; Liu et al., 2020b). Here, we apply an explicit aerosol correction to TROPOMI $NO_2$ observations near fires and explore how this changes the AMFs and a

subsequent comparison with aircraft measurements. To our knowledge knowledge this is the first comparison which focuses



on the impact of an implicit versus explicit aerosol correction of TROPOMI $NO_2$ VCDs near wildfires.

While recently, TROPOMI-derived $NO_x$ emissions have been reported (Jin et al., 2021), this study explores the derivation of top-down $NO_x$ emissions from wildfires using TROPOMI $NO_2$ observations and assesses its accuracies, with a focus on
(1) the methods used for the emission estimates, (2) the he conversion of retrieved $NO_2$ to estimates of $NO_x$, and (3) the explicit aerosol correction. We apply two methods commonly used for satellite emission estimates: (1) a flux method as previously used by, e.g., Mebust et al. (2011); Adams et al. (2019); and (2) a 3D exponential modified fit similar to that used by Fioletov et al. (2015) and Dammers et al. (2019). These two methods are applied to synthetic satellite observations with known emissions to determine the accuracy of these two methods and to explore the impact different parameters have on
the accuracy of the estimate, including sampling, noise, wind direction and speed. The $NO_2$ to $NO_x$ conversion is explored with model output and aircraft observations. Lastly, we compare the TROPOMI $NO_2$ vertical column densities (VCDs) and emission estimates to those obtained by four different aircraft campaigns in the Western United States and Canada during the 2018 and 2019 summers: (1) the Environment and Climate Change Canada's 2018 aircraft campaign over the Athabasca Oil Sand Region (AOSR) (Griffin et al., 2019; Ditto et al., 2021; McLagan et al., 2021), (2) the Western-Wildfire Experiment for
Cloud Chemistry, Aerosol Absorption and Nitrogen (WE-CAN; https://www.eol.ucar.edu/field_projects/we-can; last accessed: 19 July 2021) campaign, (3) the Biomass Burning Fluxes of Trace Gases and Aerosols (BB-FLUX) campaign (Theys et al., 2020; Kille et al., 2021), and (4) the Fire Influence on Regional to Global Environments Experiment - Air Quality (FIREX-AQ; https://www.esrl.noaa.gov/csd/projects/firex-aq/; last accessed: 19 July 2021) campaign.

This paper is structured as follows: Section 2 describes the data sets used. The emission estimation methods and the AMF estimate are described in Sect. 3. The sensitivity tests of these methods are presented in Sect. 4. An extensive comparison between the satellite observations and the aircraft measurements is detailed and discussed in Sect. 5, followed by a summary and conclusions in Sect. 6.

## 2   Data sets

### 2.1   TROPOMI

The TROPOMI instrument, the single payload on the S-5P satellite, was launched on October 13, 2017. The satellite has a Sun-synchronous orbit with a local overpass time of around 1:30pm and near full-surface coverage on a daily basis (Veefkind et al., 2012; Hu et al., 2018). The instrument's four spectrometers cover the solar spectrum in the ultraviolet (UV), near-infrared (NIR), and the short-wave infra-red (SWIR). TROPOMI, for species retrieved in the UV region, has an unprecedented high
horizontal resolution of $3.5\,km \times 5.5$ km ($3.5\,km \times 5.5$ km prior to 6 August 2019). TROPOMI $NO_2$ columns are derived from the UV-NIR spectrometer in the wavelength range of 405–465 nm. The TROPOMI standard $NO_2$ product was developed by the Royal Netherlands Meteorological Institute (KNMI) and is based on the $NO_2$ DOMINO retrieval previously used for OMI





spectra (Boersma et al., 2011); further details can be found in van Geffen et al. (2018).

Tropospheric $NO_2$ VCDs, measured by TROPOMI, represent the $NO_2$ molecules per unit area between the surface and the tropopause (in units of mol/m$^2$). These tropospheric $NO_2$ VCDs are estimated by a three step approach: (1) slant column den-

sities (SCDs, in units of mol/m$^2$) are retrieved from the spectra using a Differential Optical Absorption Spectroscopy (DOAS; Platt and Stutz (2008)), (2) the stratospheric contribution is separated, using a chemistry transport model (Boersma et al., 2004) from the SCDs to obtain a tropospheric SCDs, and (3) the tropospheric SCDs are converted to tropospheric VCDs by applying an AMF (unitless). The AMFs are estimated from a radiative transfer model (Doubling-Adding KNMI; DAK; de Haan et al. (1987); Stammes (2001); van Geffen et al. (2018)). The radiative transfer model simulates nadir-viewing radiances and

accounts for all relevant physical processes specific to the $NO_2$ light path in the troposphere, e.g., scattering, absorption and reflection. For the standard, operational AMFs, the profile shape of the TM5 model is used (at $1 \times 1°$ resolution), and the surface albedo is derived from a monthly OMI climatology (on a $0.5 \times 0.5°$ resolution) (Apituley et al., 2017). Clouds are considered in the estimation of the AMF, as well as an implicit aerosol correction by assuming aerosols to be clouds. Here, we instead re-estimate the AMFs near fire hotspots that are influenced by smoke aerosols with an explicit aerosol correction. Liu et al.

(2020b) has shown that an aerosol correction and a high-resolution $NO_2$ a priori profile can reduce large biases between the satellite observations and ground-based measurements. For this study, we use the latest data releases; the reprocessed (RPRO; 2018) and offline (OFFL; 2019-2020) $NO_2$ VCDs, which includes v1.2.2 (RPRO 2018), v1.3.1 (June 2019) and v1.3.2 (from July 2019) (Verhoelst et al., 2021). Pixels that are fully or partially covered by clouds were filtered. Here, we used 0.5 as a cut-off for the cloud fraction (referred to in teh TROPOMI files as "cloud_fraction_crb_nitrogendioxide_window", with 0

being clear-sky, and 1 complete cloud cover), and only use observations with a quality value (as referred to in the TROPOMI file as "qa_value") >0.5, with 1 being the best quality and 0 the lowest. Note that the cloud fraction cannot distinguish between smoke and clouds, as such, smoke plumes near fires are flagged as clouds. The quality and cloud fraction filters are intentionally less stringent than typically used for studies in urban areas (quality value $\geq 0.75$, and a cloud fraction $\leq 0.3$). This is because the cloud fraction is usually greater than 0.3 near fire hotspots due to the fire smoke. Therefore, to increase the

number of observations near fire hotspots we lowered the quality threshold (e.g., see Fig. 2c). The quality of the VCDs is still ensured, as we apply corrections for smoke aerosols. The standard TROPOMI tropospheric $NO_2$ VCDs are hereafter referred to as "VCD$_{KNMI}$", and the re-estimated VCDs accounting for smoke aerosols as "VCD$_{EC}$", further details about the AMF estimation can be found in Sect. 3.1.

## 2.2 GEM-MACH

For the sensitivity test of our emission estimation methods, we utilized the $NO_x$ (=$NO_2$+NO) profiles using Environment and Climate Change Canada's (ECCC's) air quality forecast model, Global Environmental Multiscale - Modelling Air-quality and Chemistry (GEM-MACH; (Makar et al., 2015b, a)). GEM-MACH is also used operationally in ECCC's operational air quality forecast system (RAQDPS, e.g., Moran et al., 2010). GEM-MACH provides .hourly output for a North American modelling domain with a 10 km x 10km grid cell size resolution, with an internal "physics" time step of 7.5 min. The chemical components





of GEM-MACH reside as a subroutine package within the model's meteorological physics model, the latter a component of the Global Environmental Multiscale (GEM) weather forecast model (Côté et al., 1998; Girard et al., 2014). GEM-MACH contains a detailed atmospheric chemistry scheme, which includes the emission and removal processes of 42 gaseous species and 8 particle species. The model run is initialized every 12 hours, at 00 and 12 UTC. In this work, the research version of

5 GEM-MACH was used that has a $10 \times 10\,\mathrm{km}^2$ grid cell size for North American domain, and 80 vertical levels (from the surface to approximately 0.1 hPa). The GEM-MACH version used in this study used a 12-bin particle's size distribution and the aerosols are assumed to be homogeneous mixtures within GEM-MACH. Further model details can be found for example in Griffin et al. (2020b). The model input fire emissions are estimated based on hotspot location using the Canadian Forest Fire Emission Prediction System (CFFEPS v2, Chen et al. (2019)). For the sensitivity tests discussed in Sect. 4, a special model run

was performed with constant fire emissions of NO, $NO_2$ and other pollutants throughout the day. For this test simulation (in Sect. 4.2), the estimates of the elevated fire emissions at 20 UTC, 13 PDT in Western USA and Canada, were used throughout the day. This simplifies determining the accuracy of the emission estimation methods, as the input emissions are constant and known.

## 2.3 Aircraft data

To compare the TROPOMI VCDs and emission estimates we use aircraft in situ and remote sensing measurements. There are limited aircraft measurements capturing fire plumes at the same time as the TROPOMI overpasses. Hence, we use measurements collected from four different aircraft campaigns specifically targeting fire emissions and smoke plume composition between 2018 and 2019, including the 2018 ECCC aircraft campaign over the AOSR, the 2018 BB-FLUX and WE-CAN campaigns, and the 2019 FIREX-AQ campaign.



### 2.3.1 ECCC aircraft campaign over the AOSR

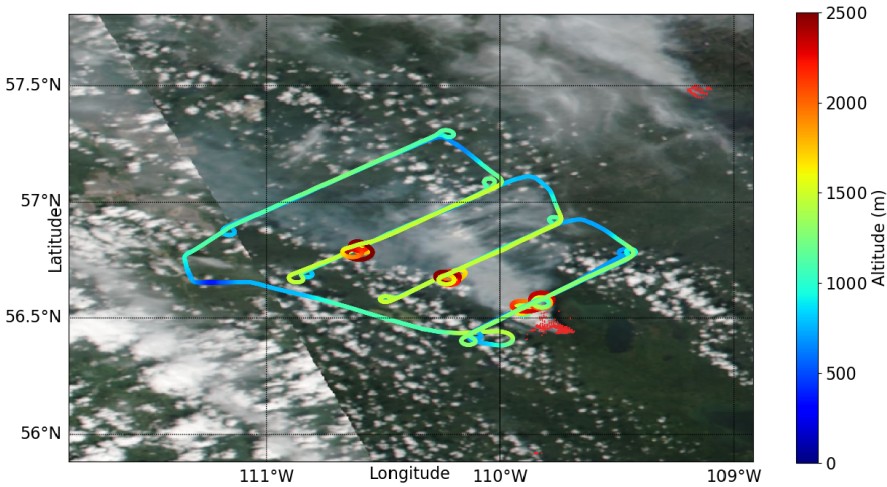

**Figure 1.** The aircraft flight tracks for the flight on 25 June, 2018 during ECCC's aircraft campaign over the AOSR are shown. The colour indicates the altitude of the aircraft. The overlay is a VIIRS true colour image together with the MODIS fire hotspots, shown as red dots (obtained from NASA Worldview; https://worldview.earthdata.nasa.gov/, last accessed: 19 July 2021).

During the ECCC's aircraft campaign over the AOSR (Griffin et al., 2019), there was an opportunity to measure downwind of a boreal forest wildfire. A large suite of measurements were taken of the Lac La Loche fire on June 25, 2018 that originated in Saskatchewan, Canada, at approximately 56°N, 110°W (Ditto et al., 2021; McLagan et al., 2021). The aircraft was equipped

with two Thermo Scientific Model 42i-TL (NO–$NO_2$–$NO_x$) analyzers, modified to measure at 1-Hz time resolution, with an uncertainty of 3 % +0.4 ppbv, and an estimated detection limit of 0.2 ppbv (Griffin et al., 2019). Note that a special photolytic converter was used to specifically measure $NO_2$, thus, the interference from other nitrogen species is null or very small.

The plume from this fire was sampled during ECCC's aircraft campaign between 20 km to 100 km downwind of the fires

and between 15:00 and 19:00 UTC (between 9:00 and 13:00 local time). Figure 1 shows the aircraft Lagrangian flight path, which sampled the same air parcels in downwind "screens" perpendicular to the wind flow direction, downwind of the source at intervals calculated from the observed winds to be separated by approximately 1 hour of advection. Each screen thus supplies a snapshot of the emissions-containing air mass, at one hour successive Lagrangian transport times downwind. This approach allows chemical transformations to be tracked in the plume following emission. perpendicular downwind of the source at

roughly the same distance apart. Multiple transects at varying altitudes were flown perpendicular to the plume direction to make up cross-plume transects at increasing downwind distances. The first transect took place between 15:00-16:15 UTC, corresponding to approximately 40 min since the time of emission based on measured wind speed and location of the source.





The second transect was sampled between 16:20 and 17:15 UTC, a cumulative time since emission of 1 h 48 min. The third transect was flown between 17:20 and 18:25 UTC, measuring the smoke plume that was emitted 2 h 32 min ago. During the fourth transect, the plume age was approximately 3 h 18 min and was sampled between 18:30 and 19:10 UTC. The pollutants measured for all four transects were emitted from the fire at approximately the same time, between 15:00 and 15:30 UTC

(based on measured wind speeds). These downwind measurements are used in this study to investigate the $NO_2$:$NO_x$ ratio and the $NO_x$ lifetime downwind of a fire plume (see Sect. 4.3). These aircraft observations could not be used to validate the TROPOMI VCDs or emission estimates as the aircraft flight took place in the morning (local time) whereas the TROPOMI overpass occurred in the afternoon when the fire was in a different burning stage. To compare the satellite VCDs and emission estimates (Sect. 5), measurements from the WE-CAN, BB-FLUX, and FIREX-AQ campaigns were used when measurements

were temporarily coincidental with the TROPOMI overpass.

### 2.3.2   BB-FLUX

The BB-FLUX campaign (https://volkamergroup.colorado.edu/timeline/field/bb-flux; last accessed: 19 July 2021; Theys et al., 2020; Kille et al., 2021) was an aircraft study conducted in the summer of 2018 in the US Northwest, based out of Boise, Idaho. The University of Wyoming King Air research aircraft (UWKA) aircraft was equipped with a Zenith Sky DOAS instrument

(CU-DOAS), measuring the UV and blue spectral ranges, and the University of Colorado Airborne Solar Occultation Flux (CU AirSOF) instrument. The aircraft flew transects underneath the plumes measuring light that passed through the smoke, thus integrating over the entire depth of the plume. CU-DOAS SCDs of $NO_2$, formaldehyde (HCHO), and nitrous acid (HONO), were observed by measuring scattered solar photons and fitted using the fitting algorithm detailed in Theys et al. (2020), with an uncertainty of 25 %. AirSOF consists of a solar tracker that keeps the instrument pointed at the sun at all times and a Fourier

Transform Infrared spectrometer to record solar spectra. Measurements in the infrared minimize Rayleigh scattering and particle extinction, and the solar tracker ensures that only photons on the direct solar beam are collected. Spectra were fit using SFit4 v0.9.4.4 to determine vertical column densities of HCHO and several other gases. The uncertainty on the HCHO retrieval is  26 % (Kille et al., 2021). AMFs for the DOAS measurements were estimated using the ratio of the DOAS-derived HCHO SCD and the AirSOF derived HCHO VCD. There is good agreement between the UV and IR cross sections (Gratien et al.,

2007), enabling the comparison of results from two spectra regions. Since $NO_2$ and HCHO are retrieved from the same DOAS fit window, the $NO_2$ SCDs can be converted to VCDs using the HCHO derived AMF.

Here, measurements from three flights that characterize two different fires are used; these three flights have good overlap with the TROPOMI overpass time (within approximately ± 30 min). The Rabbit Foot fire was measured on 12 August (RF11)

and 15 August (RF13) 2018, and originated in Idaho, US, was located at approximately 44.83°N, 114.31°W. The Watson Creek fire burned in Oregon, US at approximately 42.6°N and 120.8°W and was measured on 25 August 2018 (RF21).


### 2.3.3 WE-CAN

The WE-CAN campaign, coordinated with the BB-FLUX campaign, also took place in the summer of 2018 in Northwestern US (based in Boise, Idaho), in many cases covering the same fires as the BB-FLUX campaign (Lindaas et al., 2021). The NCAR/NSF C-130 research aircraft was equipped with numerous instruments, including a NOyO3 chemiluminescence in-

strument, which measured the NO and $NO_2$ concentrations at 1 Hz. The uncertainties are 6 % for NO and 12 % for $NO_2$ for concentrations >1 pptv. Further details about the campaign and the measurements can be found in Lindaas et al. (2021), Juncosa Calahorrano et al. (2021) and Peng et al. (2020). Data are publicly available from https://www-air.larc.nasa.gov/cgi-bin/ArcView/firexaq?MERGE=1 (last accessed: 19 July 2021).

### 2.3.4 FIREX-AQ

The FIREX-AQ campaign (Wiggins et al., 2020, ;https://csl.noaa.gov/projects/firex-aq/), sampled western U.S. wildfires onboard the NASA Douglas DC-8 research aircraft from July to August 2019. Smoke plumes were sampled with a comprehensive suite of instrumentation that measured both gas- and particle-phase species and optical properties.

NO and $NO_2$ measurements were taken with a chemiluminescence instrument; and the on-board NASA Langley Airborne

High Spectral Resolution Lidar (HSRL) (Zhou et al., 2021) measurements of aerosol extinction at 532 nm were used to calculate emissions for perpendicular plume transects as described below. The NOyO3 chemiluminescence instrument uses the same detection technique as that used during WE-CAN, and NO and $NO_2$ associated uncertainties were $\pm(5\% + 6\,\text{pptv})$ and $\pm(7\% + 20\,\text{pptv})$, respectively (Ryerson et al., 2000; Pollack et al., 2010).

Total carbon fluxes were estimated for each aircraft plume crossing using methods outlined in Stockwell (2021 in prep.). Briefly, vertical lidar aerosol extinction profiles measured on-board were scaled to total carbon using all on-board measurements of carbon-containing compounds and in situ aerosol extinction. The total carbon emission rate (g/s) was estimated by calculating a carbon flux through each pixel area, applying average wind speeds measured at several altitudes, and then integrating through the height and width of the plume. Carbon emissions were then scaled to a mass emission rate of NO

and $NO_2$ using transect-derived enhancement ratios of NO or $NO_2$ to total carbon. These ratios (specific for each transect) are applied to the carbon emissions to obtain NO and $NO_2$ emissions; the final $NO_x$ emissions are the sum of the NO and $NO_2$ emissions. The total measurement uncertainty ranged from $\sim$20-60 % by fire. In total the emissions from five different flights were within 1 h prior to the TROPOMI satellite overpass: The North Hills Fire on July 26, 2019 (46.75°N, 111.92°W, Montana, US), the Williams Flats Fire on August 3, 6 and 7, 2019 (47.94°N, 118.62°W, Washington, US), and

the Castle Fire on August 12, 2019 (36.53°N, 112.23°W, Arizona, US). Data from FIREX-AQ are publicly available and can be downloaded from https://asdc.larc.nasa.gov/project/FIREX-AQ (last accessed: 19 July 2021; DOI: 10.5067/SUBORBITAL/FIREXAQ2019/DATA001).


## 3 Methodology

### 3.1 AMF with explicit aerosol correction

Fire plumes contain significant amounts of aerosols, that scatter the UV-vis light and, thus, have a significant impact on the AMF. The standard TROPOMI $NO_2$ product does not consider aerosols but has an implicit aerosol correction by assuming

that smoke plumes are clouds. Note that smoke and clouds are distinguished by the satellite derived cloud fraction. This can introduce additional uncertainties in fire emission estimates impacted by smoke plumes. Liu et al. (2020b) found that over urban areas the implicit aerosol correction might lead to underestimated $NO_2$ VCDs of up to 50 %.

In this study, we use alternative AMFs to convert the TROPOMI SCD to a VCD and examine the impact on the TROPOMI tropospheric $NO_2$ columns near fire hotspots. This approach is very similar to previous studies focusing on the AMF estimate

(McLinden et al., 2014; Griffin et al., 2019, 2020a), with the main difference in the accounting of aerosol scatter, in the form of aerosol optical thickness (AOD). The tropospheric VCD is determined using the relationship VCD = SCD/AMF:

$$SCD = \sum_z nd(z) \cdot bAMF(z) = VCD \frac{\sum_z nd(z) \cdot bAMF(z)}{\sum_z nd(z)} = VCD \cdot AMF \quad (1)$$

Where nd(z) is the $NO_2$ number density vertical profile (in units of mol/m$^3$) along horizontal layers $z$, and $bAMF(z)$ is the layer indexed AMF; and thus the total-column AMF is defined as:

$$AMF = \frac{\sum_z nd(z) \cdot bAMF(z)}{\sum_z nd(z)} \quad (2)$$

This is summed over altitudes between the surface and the tropopause. For the plume shape of $NO_2$ nd(z), we separate the satellite observations into areas inside and outside the fire plume. Here, we use $1 \times 10^{15}$ molec/cm$^2$ (of the $VCD_{KNMI}$) as a threshold for enhanced columns, observations below this threshold are assumed to be outside the plume. Inside the plume we use a $NO_2$ a priori profile that is constant between the surface and 2.5 km (above ground) and then linearly decreases to 0 at

3.5 km (above ground), scaled by the standard KNMI VCDs ($VCD_{KNMI}$):

$$N(z) = n(z) \cdot (VCD_{KNMI} - VCD_{freetrop}), \quad (3)$$

where n(z) is the normalized profile shape, N(z) is the new a priori $NO_2$ profile used to estimated $nd(z)$ (in Eq. 4), and $VCD_{freetrop}$ is the VCD contribution in the free troposphere above 3.5 km. Between 3.5-12 km we use the concentrations from a monthly GEOS-Chem model run at the approximate time of the TROPOMI overpass on a $0.5° \times 0.67°$ resolution ver-

sion v8-03-01 (http://www.geos-chem.org, Bey et al., 2001; McLinden et al., 2014). We use the GEOS-Chem profile, as the free tropospheric $NO_2$ is not well represented in GEM-MACH due to missing elevated sources such as lightning and aircraft (Griffin et al., 2019).

The AMF(z) is the altitude-dependent AMF and is specific to each scene. Here, the SASKTRAN radiative transfer model

(Bourassa et al., 2008; Zawada et al., 2015; Dueck et al., 2017) has been used to generate an altitude-dependent AMF look-up





table (LUT) for clear-sky (and cloudy conditions), as a function of solar zenith angle, viewing zenith angle, relative azimuth angle, surface pressure, surface albedo (cloud pressure), as well an AOD (for several values between 0 and 1). For simplicity, the aerosol profile is assumed to be constant between the surface and 2.5 km (above ground) and decreases linearly to 0 at 3.5 km (similar to the a priori $NO_2$ profile used), this is consistent with typical plume heights found by Griffin et al. (2020b).

For the LUT a log-normal aerosol size distribution is assumed with $r = 0.1\,\mu m$, and $\sigma = 0.3$, and a refractive index of $1.5 + 0.1i$ at $440\,nm$ (Kou, 1996). Ozone ($O_3$) is not considered as it is not important in the wavelength range used for the $NO_2$ retrieval (van Geffen et al., 2018). Two different AMFs are estimated from the LUT, one for clear-sky ($AMF_{cs}$) and one for cloudy-sky ($AMF_{cd}$). The final AMF is estimated by using the cloud radiance fraction, $cf$ (from original TROPOMI file):

$$AMF = cf \cdot AMF_{cd} + (1 - cf) \cdot AMF_{cs} \qquad (4)$$

The cloud and clear-sky AMFs are only considered outside the plume. Inside the plume, the aerosols are already accounted for, thus, if clouds are considered again, these smoke aerosols would be accounted for twice, explicitly and implicitly (as smoke is mistaken for clouds). As such, inside the smoke plume (with the $VCD_{KMNI} > 1 \times 10^{15}$ molec/cm$^2$) we assume clear-sky ($cf = 0$) and only correct for the smoke aerosols without the additional clouds. Outside the smoke plume, the cloud fraction is taken into account, as done in the original TROPOMI AMFs, and for the cloudy-sky AMF ($AMF_{cd}$), the cloud input is taken

from the original TROPOMI files ("cloud_fraction_crb_nitrogendioxide_window", "cloud_pressure_crb").

Similar to Adams et al. (2019), we use a proxy for smoke AOD using a relationship derived between VCD enhancements and the MODIS AOD:

$$AOD = (VCD_{KNMI} - VCD_0) \times \alpha; \qquad (5)$$

with $VCD_0 = 1 \times 10^{15}$ molec/cm$^2$ and $\alpha = 3e\text{-}17\,cm^2$/molec (Bousserez et al., 2009). We have attempted to use MODIS AOD directly, however, the coverage is not sufficiently continuous to allow spatial interpolation of the daily MODIS AOD (as it changes quickly near fire hotspots). We found using the VCD enhancement as a proxy for AOD as in Eq. 5 works better. Currently, a TROPOMI AOD product is under development (personal communication with Martin de Graaf), and could be used for estimating aerosol corrected $NO_2$ VCDs near fires in the future.

Following the approach from Griffin et al. (2019), the surface pressure input is taken from the operational GEM weather forecast model, interpolated to the location and time of the TROPOMI overpass. To improve the albedo spatial resolution, we use the MODIS albedo at a resolution of $0.05 \times 0.05°$ (collection 6.1 MCD43C3; Schaaf et al. (2002)). A monthly-mean albedo is computed from the MCD43C3 files considering only 100 % snow-free pixels, snow-covered pixels are not a concern in this

study, as there are no snow covered areas near forest fires.

Outside of the plume we use the $NO_2$ profile from GEOS-Chem, and use the cloud fraction to determine the contribution between the cloudy and clear-sky AMF. The KNMI and EC VCDs are compared in Sect. 5.1. For the VCDs outside of the





plume, we found that there is very little difference between the two versions.

An example of the NO$_2$ tropospheric VCDs, with and without an explicit aerosol correction, is shown in Fig. 2. The example is from a small fire in a boreal forest in Saskatchewan, Canada (approximately 57.5°N, 109°W). Figure 2 (a) displays the
Visible Infrared Imaging Radiometer Suite (VIIRS) at approximately the same time as the TROPOMI overpass together with the MODIS thermal anomalies (red dots), showing no clouds over the fire plume. Figure 2 (b) shows the AOD used for the alternative AMFs (based on Eq. 4). The cloud fraction can be seen in panel (c), showing that the smoke plume is identified as clouds. The NO$_2$ VCD$_{KNMI}$, and VCD$_{EC}$ are shown in panels (d), and (e), respectively. This illustrates that the NO$_2$ VCD can change significantly when the AOD is accounted for, in this example, the NO$_2$ VCDs decrease nearly $2\times10^{15}$ molec/cm$^2$
over the fire hotspot (about a factor of two decrease). Note that the explicit aerosol correction can increase or decrease the VCDs, the relationship is not a simple linear relationship, it depends on the viewing geometry and AOD. In this example, accounting for clouds in addition to the smoke aerosols is probably incorrect, as there were no clouds mixed with smoke. Figure 2 (f) shows the VCDs if both the aerosols and cloud fraction are considered in the estimate: this increases the NO$_2$ VCD (in this case, again this can go either way depending on the viewing geometry and AOD) in comparison to assuming no clouds,
as in Fig. 2 (e). Panel (f) is only shown for comparison purposes, this approach accounts for smoke aerosols twice, explicitly and implicitly, and is therefore not recommended. Outside fire plumes, where the NO$_2$ is at background levels, as expected, the VCD$_{KNMI}$ and VCD$_{EC}$ are very similar.

## 3.2  Methods for estimating emissions from satellite data

Satellite observations provide information on the total amount of a trace gas released from a source, however, additional infor-
mation on transport and chemical processes is required to estimate emission rates. An important component that enables the estimation of emissions from satellite observations is information on wind direction and wind speed.

In a first step, the satellite observations are rotated to obtain an upwind-downwind domain near the emission source using the wind fields at the time and location of the observations (e.g., Pommier et al., 2013; Fioletov et al., 2015; Dammers et al.,
2019). Here, we utilize the wind fields (U, V) from the European Centre for Medium-Range Weather Forecasts (ECMWF) ERA5 dataset at a resolution of $0.25° \times 0.25°$ with an hourly output, between 1000 and 300 hPa at a resolution of 50 hPa. Note that the observations are rotated around a single point, which will cause some imperfections for large fires that are not true point sources as they are spread over larger areas.

As fire emissions can be injected into higher altitudes, wind speeds and wind directions can vary significantly at different altitudes. Griffin et al. (2020b) found that TROPOMI plume heights are a good proxy for the average height of the fire plumes. Here, we use the average TROPOMI aerosol layer heights (AER_LH) for each individual fire and use this to obtain the corresponding wind direction and speed, and average the wind fields within $\pm 50$ hPa for the corresponding plume height. In cases, where no good quality plume heights were found, we use the average plume height of fires, 2 km (or 800 hPa) (Griffin et al.,

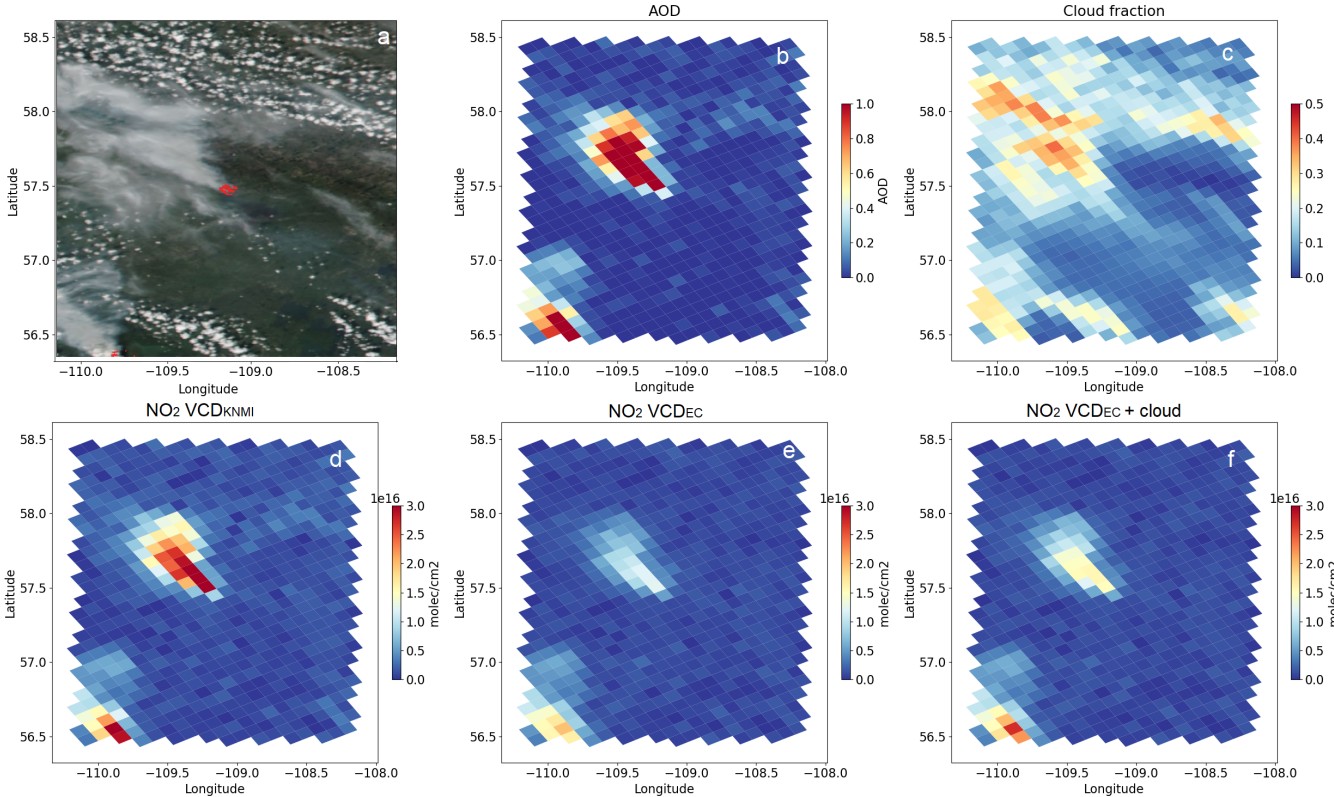

**Figure 2.** Boreal forest fire on June 25, 2018, at approximately 57.5°N, 109°W in Saskatchewan, Canada. Panel (a) shows the VIIRS true colour image together with the MODIS fire hotspots (obtained from NASA Worldview; https://worldview.earthdata.nasa.gov/). Panel (b) shows the AOD based on Eq. 5. The cloud fraction is displayed in panel (c). The tropospheric $NO_2$ $VCDs_{KNMI}$ are displayed in panel (d). And the $VCDs_{EC}$ in panel (e) and (f), without and with accounting for clouds.

2020b). The wind fields are linearly interpolated to the time of the satellite overpass.

There are multiple ways to estimate emissions from satellite observations. Here, we compare two common direct estimation methods that are best suited to estimate daily fire emissions from TROPOMI: (1) a flux method that has previously been used to estimate fire emissions from OMI (Mebust et al., 2011; Adams et al., 2019) and for $CH_4$ emission estimates using GHGSat and TROPOMI (Varon et al., 2018, 2019); and (2) a 3-D Exponentially Modified Gaussian (EMG) method (e.g., Fioletov et al., 2015; Dammers et al., 2019). A study by Jin et al. (2021) recently reported TROPOMI derived $NO_x$ emissions using a 3-D EMG method, fitting the plume in an across-wind direction.





### 3.2.1 Flux method

The flux method, also know as integrated mass enhancement method, is similar to the method used by Mebust et al. (2011), Adams et al. (2019), and (Varon et al., 2019): wind-rotated VCDs are integrated to find the total mass inside a box and account for the total mass that has entered the box. As a first step, the background is subtracted from the VCDs, this is an important
step that can influence the emission rate significantly based on the methods chosen. We investigated various ways to subtract the background: (1) $10^{th}$ percentile in the surrounding area within 100 km distance from the fire, (2) fitted background, and (3) within 25 to 50 km upwind of the fire. Based on tests with model VCDs (see Sect. 4.2, we chose to define the background based on the average upwind concentrations (method 3). Next, the VCDs are rotated around the centre of the fire, and the wind-rotated VCDs are gridded using the satellite footprint. The VCDs are then integrated inside boxes that are 4 km long
(upwind/downwind direction) and 50 km wide (perpendicular to the wind direction) and multiplied by the wind speed to find $E_y$, the flux (in g/s) y km downwind of the fire. The initial emissions at the source as (in detail described in Mebust et al. (2011)) are found by:

$$E = E_y \cdot \frac{t_c}{\tau \cdot (1 - exp(\frac{t_c}{\tau}))} \tag{6}$$

$t_c = x_c/u$ is the residence time inside the box of a width $x_c$ and wind speed $u$, and $\tau$ is the lifetime or e-folding time. This
method is very sensitive to the wind speed as it directly impacts the emission rate. The final emission rate obtained using this method is an average of all emission rates within 20 km downwind of the fire centre. An example is shown in Fig. 3 for the Watson Creek fire (also observed by the BB-FLUX campaign) on August 25, 2018 at approximately 43.5°N, 120.7°W in Oregon, US. Panel (a) shows the raw $NO_2$ VCD as observed by the satellite with the fire hotspot in the middle. Panel (b) shows the wind-rotated and smoothed VCDs (background subtracted), here, the red lines show the area of the $4 \times 50$ km$^2$ boxes
used for the estimate of the emission flux. The emissions (following Eq. 6) for each box are shown in panel (c). To obtain the final number for the emissions only the boxes within 20 km of the fire are averaged (Adams et al., 2019), which ensures that the entire fire is captured. The first box, near the fire hotspot, is larger to make sure that the entire fire area is captured which might be a few kilometers upwind of the fire centre. Further than 20 km downwind of the fire the uncorrected emission rate (black stars) drops due to the short lifetime (including chemistry, deposition and dispersion) of $NO_2$. The colours in Fig. 3 (c)
indicate the different assumed lifetimes ($\tau$) for $NO_2$; 1 h (red), 2 h (green); 4 h (blue), and 6 h (purple), and no correction of the lifetime (equivalent to infinite lifetime; black stars). The differences in the inferred emissions for different $NO_2$ lifetimes ($\tau$) are relatively small if the lifetime is longer than 3 h, however, the impact on the retrieved emissions increases rapidly as $\tau$ falls below two hours.

### 3.2.2 Exponentially Modified Gaussian

While the flux method simply sums up all mass emitted by the fire, emissions can also be estimated by fitting a Gaussian plume to the observations. To describe the distribution of the $NO_2$ VCD field near the source, an exponentially modified Gaussian (EMG) function can be used (see Eq. A1-7, in the Appendix). The EMG method has previously been applied to estimate $SO_2$

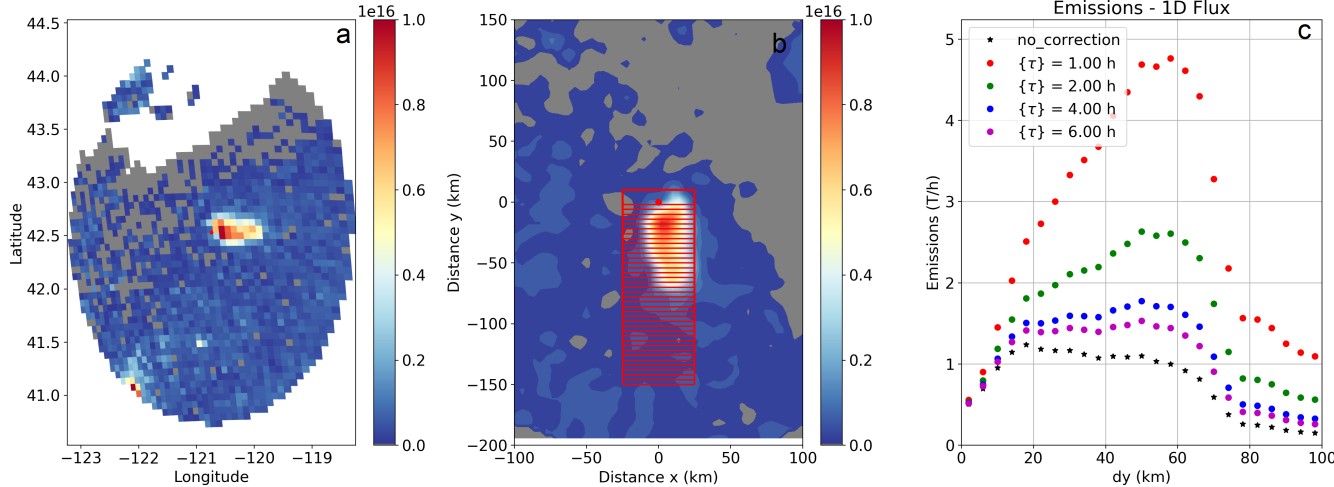

**Figure 3.** Example of the flux fit for the Watson Creek fire on August 25, 2018 at approximately 43.5°N, 120.7°W in Oregon, US. The TROPOMI observations ($VCD_{EC}$) are shown in panel (a) in a latitude/longitude domain with the fire centre in the middle (shown as a red dot). The background is subtracted from the VCDs, and then rotated (along the wind direction) and smoothed (weighted by the footprint coverage) shown in panel (b), in an upwind-downwind domain (in km) with the fire at its centre. The mass is summed inside the $50\text{x}4\,\text{km}^2$ (shown as red boxes) and using Eq. 6 the emission rate is estimated for each of the boxes assuming a variety of lifetimes for $NO_2$, shown in panel (c).

(Fioletov et al., 2015; McLinden et al., 2020) and $NH_3$ (Dammers et al., 2019) emissions from satellite observations. Here, contrary to the previous studies that used many days or even years to estimate the emissions, the observations are not gridded by different wind speeds as only single days are fitted. After applying a wind-rotation to the tropospheric VCDs, the EMG was used to estimate the emissions from a point source. The lifetime ($\tau = 1/\lambda$) and the plume spread ($\sigma$) can be estimated at the same time, however, there are many solutions for $\lambda$ and $\sigma$. Therefore, to avoid over-fitting the parameters due to the limited amount of observations for single days, $\lambda$ and $\sigma$ were kept constant. Natural variation of $\lambda$ and $\sigma$ are later accounted for in the uncertainty estimate.

An example of the method is shown for the Watson Creek fire on August 25, 2018 in Oregon, US. Figures 4 (a) and (b) show the satellite observations in a longitude/latitude domain and in a wind-rotated upwind/downwind domain, respectively. The lower panels (c) and (d) show the fitted VCDs (after the EMG has been applied) in the longitude/latitude domain and the upwind/downwind domain, respectively.







**Figure 4.** Example of the EMG method for the Watson Creek fire on August 25, 2018 at approximately 43.5°N, 120.7°W in Oregon, US (same as in Fig. 3). The TROPOMI NO₂ observations (VCD$_{EC}$) are shown in a latitude/longitude and upwind/downwind domain in panels (a) and (b), respectively. The fitted VCDs using the EMG are shown in panels (c) and (d) in the longitude/latitude and upwind/downwind domain, respectively.



## 4  Accuracy of the emission estimates using synthetic data

To determine the accuracy of the emission estimates, we use synthetic $NO_2$ and NO VCDs with prescribed emissions and test if these emissions can be determined with the flux and EMG methods, as described in the previous section. The GEM-MACH air quality model was used to obtain the synthetic VCDs. Here, we use a special model run, where the emissions from various fire

hotspots are held constant for a 24 h period to remove any diurnal variability. This is needed in order to simplify the sensitivity study and to determine if the methods can accurately reproduce the input emissions, as any diurnal variability will impact the VCDs over time (downwind), and thus, complicate the analysis. The model $NO_2$ and NO profiles are integrated over the first 39 layers (approximately 10 km) to obtain VCDs, the $NO_x$ VCDs are the sum of the $NO_2$ and NO VCDs. $NO_2$ VCDs cannot be directly compared to $NO_2$ input emissions due to the GEM-MACH model chemistry and oxidation from NO to $NO_2$. Thus,

we use $NO_x$ VCDs and compare those to the $NO_x$ model input emissions (ENO+$ENO_2$). In the model approximately 90 % of $NO_x$ is emitted as NO and converts to $NO_2$ (based on the reaction mechanism within the model). In this section sensitivity tests are performed testing the flux and EMG methods using model $NO_x$ VCDs and known emissions. Since TROPOMI can only measure $NO_2$, the scaling from $NO_2$ to $NO_x$ is important for the emission estimate and discussed in the following section section.

### 15   4.1  Lifetime and plume spread

The $NO_2$ lifetime (or decay time) and plume spread (or dilution) can be determined with the EMG method (see Eq. A1-7). However, based on our analysis, using just single overpasses these only return reasonable results of the lifetime and plume spread for less than 30 % of fires. Thus, for this study we kept the lifetime and plume spread the same for each fire. A variety of fires were used to determine a suitable lifetime and plume spread using the EMG method. Based on good EMG fits for various

fires, we obtained a mean lifetime of 1 h ($\pm$0.5 h) for $NO_x$ and a plume spread of 6 km ($\pm$1 km) when the model VCDs are used. When applying the EMG method to TROPOMI observations, we derived a mean lifetime of 2 h ($\pm$1 h) for $NO_2$ and a plume spread of 7 km ($\pm$1 km). Note that the difference of the lifetime between the model and the TROPOMI observations are expected, since the chemical lifetime of $NO_2$ is shorter in the model compared to reality (it can be seen in the fire plumes that dissipate faster in the model compared to the satellite observations). The lifetime derived from the EMG is not a true chemical

lifetime, but is also influenced by plume dispersion and surface deposition as described by de Foy et al. (2015). Juncosa Cala-horrano et al. (2021) found an average $NO_x$ lifetime or e-folding time of 90 min inside fire plumes using aircraft measurements during the WE-CAN campaign. Our satellite-derived lifetime of 2 h ($\pm$1 h) using the EMG method agrees with their results within the uncertainties. The plume spread parameter incorporates several effects, including the diffusion of the plume in the cross-wind-direction, the spatial extent of the source, and the size of the satellite pixel. The plume spread parameter is only

used for the EMG method, and the flux method does not take this into account. Note that for the EMG, changes in lifetime and plume spread can compensate for each other: a shorter lifetime will increase the emissions and a smaller plume spread will decrease the emissions. Thus, the emissions are almost identical (within 5-10 %) when using for example $\sigma = 7$ km, and





$\tau = 1.5\,\mathrm{h}$, and $\sigma = 6\,\mathrm{km}$ and $\tau = 1\,\mathrm{h}$.

For the estimates in this section, when using the model VCDs, we apply a constant plume spread and lifetime of 6 km and 1 h. When utilizing the TROPOMI observation (in Sect. 5), we set the lifetime, for both EMG and flux method to $\tau = 2(\pm 1)\,\mathrm{h}$.

## 4.2 Reproducing the synthetic emissions

In this section, four sensitivity tests are performed testing the flux and EMG method (as described in Sects. 3.2.1 and 3.2.2): (i) using the model sampling and model winds (best case scenario), (ii) using satellite sampling, (iii) using satellite sampling and ERA winds (different winds than the winds used in the GEM-MACH model), and (iv) using satellite sampling and ERA winds, and adding a random error similar to that of the TROPOMI observations (scenario closest to using satellite observations). The results from the sensitivity test are shown in Fig. 5 for the flux method and EMG method, respectively. In total the emissions of over 59 fires (in North America), for the month of June 2018 were successfully retrieved and subsequently compared. Two fires had unusual wind conditions, with very high winds and wind-shear that have been excluded from the analysis. Scenario (i) is shown in panel (a), scenario (ii) in panel (b), scenario (iii) in panel (c), and scenario (iv) in panel (d).

For scenario (i) the emissions are highly correlated to the input emissions with $R > 0.8$ for both the flux method and the EMG. The fitted emissions are biased high by about 37 % (based on the slope) using the flux method. The EMG is more accurate for this scenario and has no bias with a line of best fit close to the 1:1 line.

Scenario (ii) assumes satellite sampling (the synthetic observations are filtered when the real TROPOMI quality flags are less than 0.5), thus, the number of observations for the fit will be less, especially close to fire where observations are removed by the cloud filter due to the high smoke content. The impact of this on the emission estimate is shown in panel (b) in Fig. 5. This impacts the number of fires that can be retrieved: reduced to 53 fires, roughly 10 % fewer successful fire retrievals can be expected, this is the case for both methods. Furthermore, using the satellite sampling leads to, on average, lower emissions: the flux method, based on the slope, is still biased high by about 15 % and the EMG is now biased low by about 23 % (the relative difference is about 10 %) compared to the "true" synthetic emissions. The correlation coefficient is slightly smaller, but still shows a correlation with $R \sim 0.9$ for both methods.

In scenario (iii), see panel (c) in Fig. 5, in addition to the satellite sampling the winds were changed to the wind fields from ERA5, which can be different than the wind fields of the model. Since in reality the true winds are unknown and likely differ to some extent from the ERA5 winds, this scenario is a more realistic scenario compared to the two previous ones. This has little impact on the slope and correlation.

The measured VCDs are not perfect and have some instrument noise. In the previous tests, perfect model VCDs were used, whereas in this last scenario (iv), a random error has been applied to the model VCDs. A random error of $0.7 \times 10^{15}$ molec/cm$^2$





was applied to the model VCDs, and is similar to the reported noise of the TROPOMI observations. Adding a random error has a minimal effect on either the flux or EMG method, see panel (d) in Fig. 5.

5    While the slope is closer to the 1:1 line for the flux method compared to the EMG method, for scenario (ii)-(iv), there is less scatter and more consistency for the EMG method especially for emissions less than $5\,\mathrm{t[NO]/h}$ (which the range of the emissions compared in in Sect. 5). For instance, for emissions less than 5t[NO]/h, for scenario (iv), the EMG has a relative difference of $-3\,\%$ and a slope of 1.24, whereas the flux method has a relative difference of $-42\,\%$ and a slope of 1.94.

**Figure 5.** The results of the sensitivity test with synthetic data for test (i)-(iv) are illustrated (see text for detailed description of the scenarios). The fitted emissions applying the flux (orange triangles) method and the EMG method (blue downward triangles) versus the model input emissions are plotted together with the statistics (slope of best-fit using the geometric mean, s; correlation coefficient, R; the number of points, n; and the mean and standard deviation of the relative difference, rel. Diff: input−fitted).





### 4.3 NO$_2$ to NO$_x$ scaling

The chemistry of NO$_x$ is complex including a fast inter-conversion between primary emissions of NO and secondary NO$_2$. Current satellites, such as TROPOMI, can only measure NO$_2$. Thus, the scaling factor from NO$_2$ to NO$_x$ is important. Only a limited number of studies investigate this scaling for satellite derived NO$_x$ emissions from NO$_2$ observations (e.g., Adams et al., 2019; Lorente et al., 2019). Here, we use the synthetic data and derive NO$_x$ emissions from NO$_x$ VCDs and compare these to NO$_2$ emissions from NO$_2$ VCDs by applying the EMG to those VCDs. This can help to understand how the satellite-derived NO$_2$ emissions can be scaled to NO$_x$ emissions and if this is even possible; i.e. a large scatter of NO$_x$ (from NO$_x$ VCDs) and NO$_2$ (from NO$_2$ VCDs) derived emissions would indicate that the conversion is not stable and NO$_x$ emissions could not be derived from NO$_2$ VCDs without further information of other parameters. The results are shown in Fig. 6, where there is a perfect correlation ($R = 1$) between the fitted NO$_2$ and NO$_x$ emissions that are based on the model VCDs for 59 different fires across North America. While the NO$_x$ input emissions are emitted as 90 % NO and 10 % NO$_2$ for all fires the conversion and lifetime can change based on the different OH, NO$_x$ concentrations, and temperature. Note that for this fit, all VCDs between 25 km upwind and 100 km downwind are used where the NO$_2$:NO$_x$ ratio is changing with plume age. The derived ratio of 0.68, that allows to convert the derived NO$_2$ emissions (from NO$_2$ VCDs) to total NO$_x$ emissions, has a perfect correlation indicating that the scaling from derived NO$_2$ emissions to the net NO$_x$ emissions is stable. This derived ratio has been applied to the TROPOMI-derived NO$_2$ emissions in this study to convert these to total NO$_x$ emissions. Note that the model VCDs used for this analysis are from 20 UTC, close to the TROPOMI overpass time, a different ratio is likely for other times of day. We note that the emissions of NO$_x$ will largely be in the form of NO: our NO$_2$ to NO$_x$ ratios above serve to convert our measured quantity, a satellite-derived "emission" of NO$_2$, to the net quantity relevant for emissions inventories and modelling, the total emissions of NO$_x$. The ratio described here is intended as a correction to return the net emissions of NO$_x$, and should not be interpreted as the ratio of NO$_2$ to NO$_x$ during the actual emissions process itself.

To further support the NO$_2$:NO$_x$ scaling, we also looked at aircraft measurements taken during the ECCC's aircraft campaign over the AOSR (described in Sect. 2.3.1), on June 25, 2018 in Saskatchewan, Canada and compared those to the model output for the same fire. The aircraft measurements were taken from near the surface to the top of the fire plume, at four downwind cross-plume transects at distances of approximately 20 km to 100 km from the wildfire. The NO$_2$:NO$_x$ concentration ratios were found from correlation of the scatter plots; the slope and the slope error (as error bars) from the aircraft measurements are shown in Fig. 7 (b), for comparison, the model NO$_2$:NO$_x$ for that fire are shown in Fig. 7 (a). The NO$_2$:NO$_x$ slopes from the aircraft measurements have very high correlations for all four flight transects with $R^2 > 0.8$ ($R^2 = 0.96$, $R^2 = 0.9$, $R^2 = 0.86$, and $R^2 = 0.81$ for transect 1, 2, 3, and 4, respectively). Near the fire the ratio is 0.71($\pm$0.03) and is consistent with the model derived ratio, as shown in Fig. 6. Downwind of the fire the ratio increases as more NO is oxidized to NO$_2$, this can be seen in the model results as well. The model NO$_2$:NO$_x$ has the same intercept as the measured ratio, only the slope is lower indicating the NO does not oxidize to NO$_2$ fast enough in the model, this leads to a lower NO$_2$:NO$_x$ ratio further downwind (Fig. 7). It should be noted that for the EMG and thus the conversion from NO$_2$ to NO$_x$ emissions, the VCDs close to the hotspot (roughly within



20 km from the fire) are driving the emission estimate. The ratio further downwind does not significantly impact the emission estimate, even though at 100 km downwind almost all measured $NO_x$ is $NO_2$, because these VCDs are not driving the emission estimate and this ratio is not significant for the $NO_2$ to $NO_x$ scaling of the satellite-derived emissions. Close to the fire (within 20 km of the centre) the aircraft observations show a $NO_2:NO_x$ ratio of 0.71-0.75. This analysis shows that the derived $NO_2$

5   emissions can be scaled to $NO_x$ emissions, and that this conversion is not the most significant source of uncertainty. It should be noted that this might be different for mountainous areas or large fires where the plume is lifted into the free troposphere, but for the fires investigated in this study the ratio to convert satellite-derived $NO_2$ emissions to $NO_x$ emissions is stable. The model output is in good agreement with the aircraft observation which show a similar $NO_2:NO_x$ ratio close to the fire. For studies looking to convert satellite derived $NO_2$ emissions to $NO_x$ emissions, based on this analysis, we recommend using a

10   value between 0.68-0.75 for day time satellite derived $NO_2$ emissions (for early afternoon overpasses). For this study, we use a ratio of 1/0.68 for the scaling from $NO_2$ to $NO_x$ emissions. We also conducted further tests by converting TROPOMI $NO_2$ VCDs to $NO_x$ VCDs, using a different ratio inside and outside the plume, to then determine the $NO_x$ emissions. However, we found that this introduced more uncertainty and scatter in the emission estimate. Based on these tests, we recommend retrieving the $NO_2$ emissions from satellite $NO_2$ VCDs before converting the derived $NO_2$ emissions to $NO_x$ emissions.

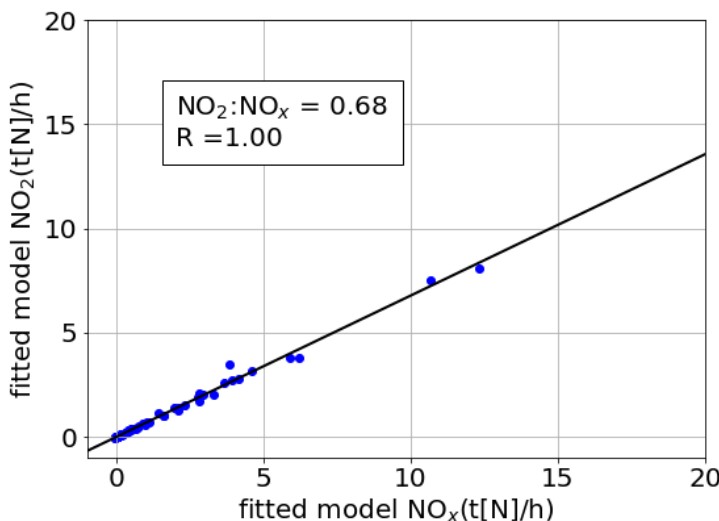

**Figure 6.** Fitted emissions derived using the EMG method for the synthetic $NO_2$ and $NO_x$ ($NO_2$+NO) VCDs, suggesting a $NO_2:NO_x$ ratio of 0.68 should be used for the conversion, the fitted $NO_2$ and $NO_x$ emissions are perfectly correlated ($R = 1$).



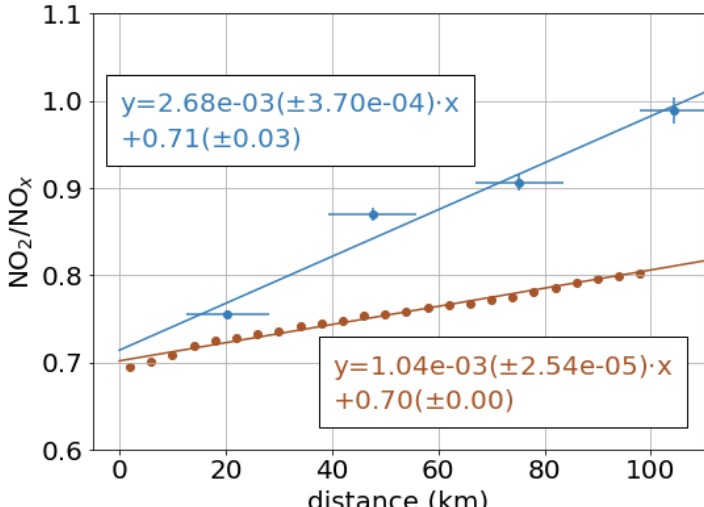

**Figure 7.** $NO_2$:$NO_x$ ratio using model output (orange) and aircraft measurements (blue) for the Lac La Loche fire on June 25, 2018. The model ratio using the synthetic $NO_2$ and $NO_x$ ($NO_2$+NO) VCDs downwind of the fire is shown as orange dots. The $NO_2$:$NO_x$ ratio using in situ aircraft measurements for the same fire is shown as blue dots. Four cross-plume transects were flown at various altitudes capturing the entire fire plume at different distance from the fire.

### 4.4 Total uncertainties of the $NO_x$ emission estimate

Overall, we found that the EMG method can accurately reproduce the model emissions exactly under perfect conditions, i.e for model sampling and model winds, scenario (i). The sampling of the satellite has very little impact on the emissions for the EMG method for typical fires, scenario (ii). The imperfect winds result in the largest uncertainty and leads to an overall low biased emission estimate. The added noise did not impact the results of the EMG method.

The flux method cannot reproduce the input emissions as well under a perfect scenario and tends to overestimate the emissions. However, the bias is reduced for the imperfect scenarios, as these uncertainties, such as the satellite sampling and ERA5 winds, overall reduce the emission estimate. Also for the flux method, the satellite noise had little impact on the emission estimate. The satellite sampling leads to approximately 10 % less successfully derived emissions for both methods, which is mostly due to cloud cover or very thick fire smoke.

In order to estimate the total uncertainties for the satellite-derived emission estimates, we consider the following uncertainties: (1) the uncertainty from the method itself, (2) the uncertainties of the satellite VCDs, (3) the $NO_2$:$NO_x$ conversion, (4) the $NO_x$ lifetime, and (5) the uncertainty of the winds. A summary of uncertainties can be found in Table 1. For the method uncertainty, we use the relative difference of scenario (ii). The uncertainty due to the wind speed and plume height is based



**Table 1.** Summary of uncertainties for the satellite emission estimates.

| Type | Uncertainty range | Uncertainty Flux method | Uncertainty EMG |
|---|---|---|---|
| Satellite VCDs | | 20 % | 20 % |
| Method | | 28 % | 9 % |
| $NO_2$:$NO_x$ | 0.68-0.75 | 6 % | 6 % |
| Lifetime | $\pm 1\,\mathrm{h}$ | 30 % | 25 % |
| Wind | GEM vs ERA5 | 26 % | 18 % |
| Total | | 53 % | 38 % |

on the tests using different winds, where we use the relative difference between the emission estimates from scenario (ii) and (iii). The $NO_2$:$NO_x$ uncertainty is based on the range of values we found for the conversion (0.68-0.75). The uncertainty due to lifetime is based on estimates using different lifetimes and plume spread ranges. The uncertainty of the satellite VCDs (20 %) is based on previous estimates of the AMF uncertainties (McLinden et al., 2014; Griffin et al., 2019), and a similar number

was also obtained by comparing the new satellite VCDs to the aircraft VCDs in Sect. 5. To obtain the overall uncertainty, we added those uncertainties in quadrature. Note that this might overestimate the actual uncertainty as these components of the net uncertainty may have compensating effects leading to a better estimate; for example, the uncertainty of the winds leads to smaller emissions for the flux method which partially compensates for the overall high bias from the method.

Comparing the flux method to the EMG method, we found that the EMG method has higher correlation coefficients, less scatter for the $NO_x$ emission estimates, and smaller total uncertainties. However, one of the primary disadvantages of the EMG method is the uncertainty in lifetime and plume spread. Thus, we would recommend a constant lifetime and plume spread when doing single day or overpass emission estimates with TROPOMI observations. It should also be noted that while the EMG successfully estimates the emissions for a short lived species like $NO_x$ this method does not work as well for longer-

lived species such as CO or $CH_4$, as these do not typically obtain a Gaussian plume shape as well (due to the long lifetime), except under very stable wind conditions. We would recommend using the flux method to obtain the emissions for those species.

## 5   Comparison to aircraft measurements

In Sect. 3.1, we described how new AMFs with an explicit aerosol correction were derived. Here, those newly estimated

TROPOMI VCDs and TROPOMI-derived $NO_x$ emissions are compared to aircraft measured VCDs and aircraft-derived emissions. We compare (1) integrated VCDs utilizing measurements from the WE-CAN and FIREX-AQ campaign, similar to the previous work of Griffin et al. (2019), (2) $NO_x$ emissions derived from airborne lidar and in situ carbon and nitrogen mea-



surements from the FIREX-AQ campaign, and (3) TROPOMI VCDs and emission estimates to aircraft remote-sensing DOAS measurements taken during BB-FLUX campaign (following the approach from Theys et al. (2020)).

## 5.1 Integrated profiles

To compare the aircraft measurements to the TROPOMI VCDs, the aircraft in situ measurements (flown as transects or spirals at various altitudes) are integrated to VCDs and averaged within the TROPOMI pixel, following the approach presented in Griffin et al. (2019), however, here we use a stricter coincident criterion of $\pm 30$ min of the TROPOMI overpass. This somewhat limits the number of measurements; however, fire emissions are highly variable and thus relaxing the coincident criterion may affect the comparison. In total 42 TROPOMI observations are compared to the aircraft measured VCDs from 12 different flights across two studies. An example profile is shown in Fig. 8 (c), where the black dots indicate the aircraft measurements and the red line is the interpolated profile used to estimate the aircraft VCD. To account for $NO_2$ measured above the aircraft, we include a monthly GEOS-Chem profile; however, this will account for very little of the total tropospheric VCD ($\sim 1 \times 10^{14}$ to $5 \times 10^{14}$ molec/cm$^2$). Below the aircraft we assume a constant volume mixing ratio (VMR) based on the measurements at the lowest aircraft altitude. The error bars shown in Fig. 8 indicate different profile extrapolation methods to the ground. On the lower end: an elevated plume is assumed and the VMR from the lowest altitude of the aircraft linearly decreases to 0 at the surface, and on the upper end: twice as much $NO_2$ as the measurement of the lowest aircraft altitude is assumed near the surface.

Figure 8 (a) and (b) show the comparison for the $NO_2$ $VCD_{KNMI}$ and $VCD_{EC}$, respectively. Based on the correlation, the slope of best fit and the mean difference between the aircraft and TROPOMI VCDs, the comparison suggests that the newly derived AMFs ($VCD_{EC}$) show an improvement over the original $VCD_{KNMI}$. Note that only a limited number of measurements are available, especially with high $NO_2$ VCDs. Thus, the slope and correlation is primarily driven by one high observation. An example of a profile (measured and interpolated) is shown in Fig. 8 (c): there are gaps in the measurements due to the stringent coincident criteria, and this comparison is not ideal. Thus, we included two further comparisons to aircraft-borne observations and aircraft-derived emissions in the following sections.





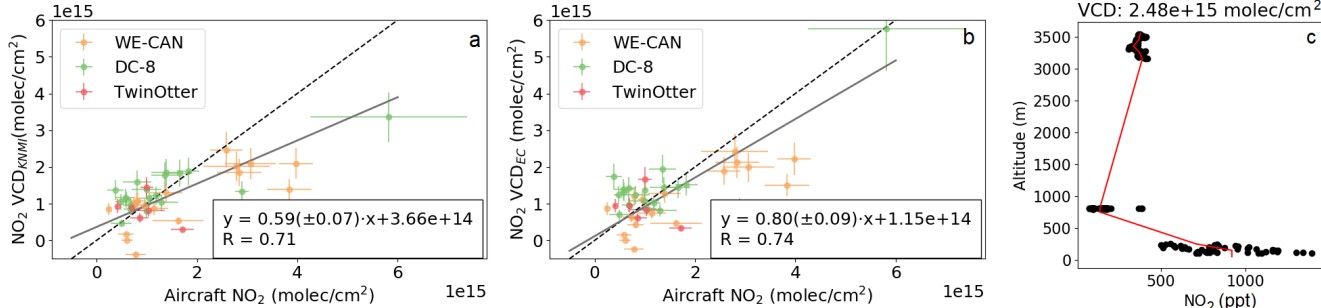

**Figure 8.** Comparison between aircraft measurements and TROPOMI NO$_2$ VCDs. Aircraft data was collected as part of the WE-CAN and FIREX-AQ (collected on the DC-8 and Twin Otter aircraft) campaigns in the western US in 2018 and 2019, respectively. The comparison is shown using (a) the VCD$_{KNMI}$ and (b) the VCD$_{EC}$. An example aircraft profile is shown in panel (c), where the black dots indicate the aircraft measurements (WE-CAN campaign, RF08) and the red line is the interpolated profile used to estimate the aircraft VCDs.

## 5.2 Emission comparisons

We compared the TROPOMI derived emissions to aircraft derived emissions from measurements taken during the FIREX-AQ campaign, as described in Sect. 2.3.4. To compare the satellite and aircraft-derived emissions, the time of the plume emission is estimated. For the aircraft emissions, the time of emission is based on the mean time, $t_t$, when the transect was flown (the transects typically take less than 5 min). We then assume, the time of emission is $t_t - \tau$ for the aircraft measured plume, where $\tau$ is the plume age. Plume ages were estimated by averaging HYSPLIT back trajectories from the aircraft position during the plume transect to the fire source using multiple meteorological datasets to account for spatial and temporal variations in the wind. Uncertainties are driven by errors in the meteorological datasets (wind variation), assumed vertical velocities, and inaccuracies in the fire source location. For the satellite observations the time of the emission is not as precise, as many measurements downwind of the fire are used for the estimate. For the flux estimate only measurements up to 20 km are used and averaged. For the EMG observations further downwind are used, however, as the magnitude of the NO$_2$ columns decreases downwind, they become less important for the overall magnitude of the enhancement $a$ (see Eq. A1). Thus the most important observations are roughly within 20 km of the source (depending on the wind speed). The time of emissions for the satellite observations (based on average wind speeds), are an average of the hour prior to the satellite overpass. We define roughly the time of emissions for the satellite observations to be 30±30 min prior to the satellite overpass. The time of emission from the satellite-derived emissions is a range of times and a precise time cannot be determined, because the satellite NO$_2$ amounts that go into the emission estimate (downwind of the fire) were emitted at various times before the satellite overpass.

The comparison between the aircraft and the satellite derived NO$_x$ emission rates for the five overlapping flights are shown in Fig. 9 using the VCD$_{EC}$ (the same is shown in Fig. B1 using the VCD$_{KNMI}$, shown in the Appendix). The magnitude of total emissions from five different flights, could be compared, where the time of emission was within 1 h prior to the satellite





overpass: The North Hills Fire, the Williams Flats Fire and the Castle Fire. As $NO_x$ has a short lifetime, the aircraft emissions were adjusted accordingly using the HYSPLIT estimated plume age (red triangles). A lifetime of 2 h (as was derived from the TROPOMI observations and EMG fits, see Sect. 4.1) was applied to the aircraft $NO_x$ emissions (by multiplying a factor of $1/exp(-plumeage/lifetime)$) to determine the initial emission rate at the time of emission from the fire, for comparison,

the original aircraft emissions are also shown (pink triangles). The fire radiative power (FRP) from the Geostationary Environmental Satellite System 17 (GOES-17) of these fires are shown in Fig. 9 as small grey dots as an indicator for diurnal fire intensity. GOES-17 is a geostationary satellite, also referred to as GOES-West, providing information such as FRP every 5-15 min primarily over the Western part of North America (Li et al., 2020, and references therein). The aircraft-derived $NO_x$ emissions follow the GOES-17 FRP well, with increased FRP tracking increased $NO_x$ emissions. Six TROPOMI overpasses

are coincident (shown as shaded grey areas in Fig. 9) with aircraft-derived emissions. The satellite- and aircraft-derived emissions are summarized in Table 2. The best agreement between the aircraft and satellite-derived emissions is found using the EMG method with the $VCD_{EC}$, for which the satellite and aircraft-derived emissions are within the estimated uncertainties except for the Williams Flats fire on 3 August, where the satellite-derived emissions are higher. For the first TROPOMI orbit on 3 August, the emissions are very low, and for the second orbit, the fire activity was increasing rapidly, which is likely why

there are discrepancies between the satellite- and aircraft-derived emissions that day. The flux method always results in smaller emissions compared to the EMG and has a low bias compared to the aircraft-derived emissions.

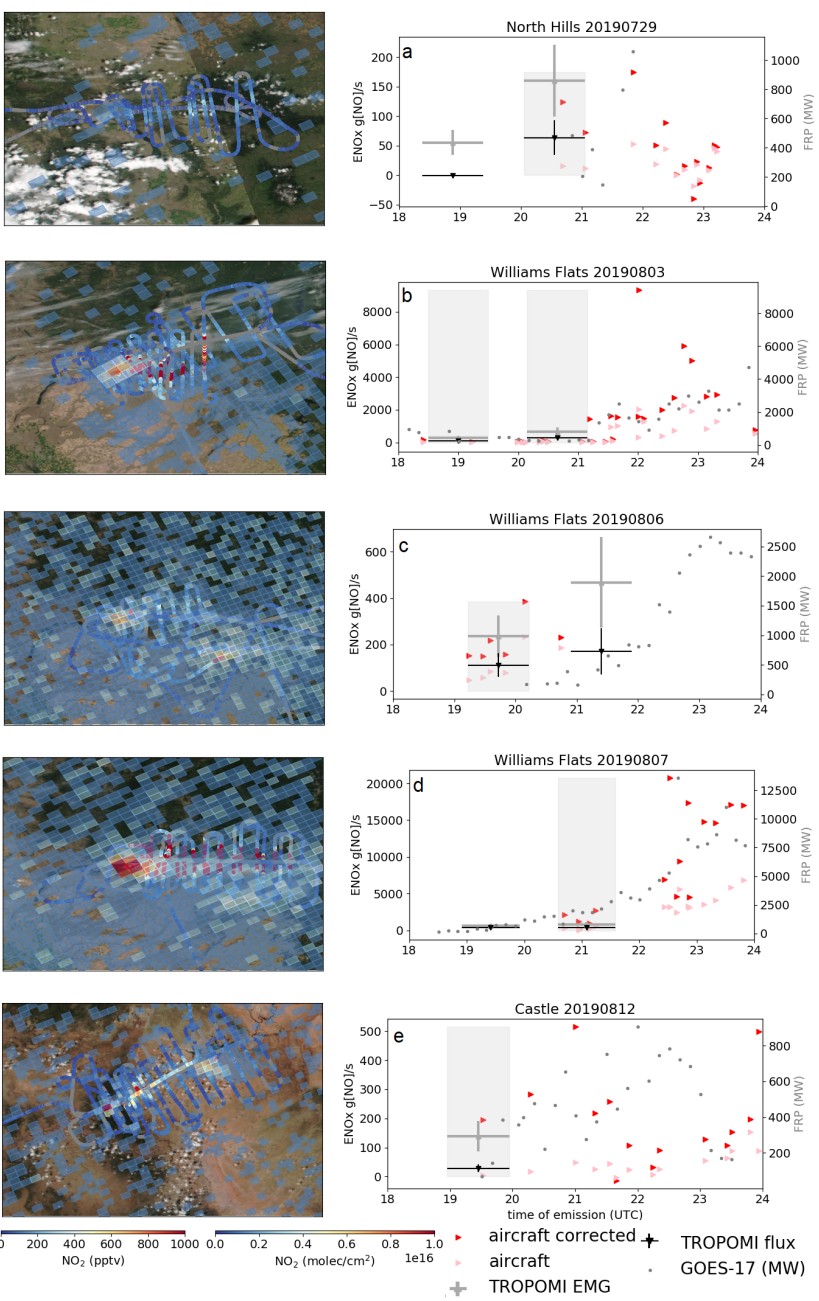

**Figure 9.** Comparison between aircraft and TROPOMI $NO_x$ emission rate estimates. Aircraft data was collected as part of the FIREX-AQ campaign on the DC-8, for five flights aircraft and TROPOMI measurements are coincident. The TROPOMI $VCD_{EC}$ (>1×10$^{15}$ molec/cm$^2$), together with the aircraft $NO_2$ (in pptv) and VIIRS overlays (obtained from NASA Worldview; https://worldview.earthdata.nasa.gov/, are shown on the left. The aircraft-derived emissions are shown as pink and red triangles. The red triangles are the aircraft-derived emissions corrected assuming a lifetime of 2 h. The TROPOMI-derived emissions are estimated with EMG (grey) and flux (black) method utilizing the $VCD_{EC}$. The grey shaded areas indicate the times when the aircraft- and satellite-derived emissions were coincident. The aircraft-derive emissions have an uncertainty of 20-60 % (not shown here) that can be seen in the spread. The GOES FRP in MW (right axis) is shown as small grey dots and indicate the change of the fire activity during the day.





**Table 2.** Summary of the satellite (using $VCD_{EC}$ for the estimate) and aircraft-derived $NO_x$ emission estimates (in t[NO]/h). The uncertainties for the satellite-derive emissions are based on Table 1, and the aircraft-derived uncertainties are approximately 40 %.

| Fire | TROPOMI EMG (t/h) | TROPOMI flux (t/h) | lifetime corr. aircraft (t/h) | aircraft (t/h) |
|---|---|---|---|---|
| North Hills (29 Jul) | 0.6±0.2 | 0.2± 0.1 | 0.5± 0.3 | 0.06±0.03 |
| Williams Flats (3 Aug, 19UTC) | 1.0±0.4 | 0.4± 0.2 | 0.2±0.1 | 0.02±0.01 |
| Williams Flats (3 Aug, 20.5UTC) | 2.4±0.9 | 1.0± 0.5 | 0.3± 0.1 | 0.08±0.03 |
| Williams Flats (6 Aug) | 0.9±0.2 | 0.4± 0.2 | 0.8± 0.5 | 0.36±0.14 |
| Williams Flats (7 Aug) | 2.9±1.1 | 1.5± 0.8 | 5.2± 3.1 | 0.97±0.39 |
| Castle (12 Aug) | 0.5±0.2 | 0.1± 0.1 | 0.7± 0.4 | 0.02±0.01 |

## 5.3 DOAS comparison

As a third comparison, we included the DOAS observations taken as part of the BB-FLUX campaign, here referred to as CU-DOAS. A total of three flights and three TROPOMI overpasses were found to be near-synchronous with good coverage of TROPOMI and aircraft measured $NO_2$. The flights measured the Rabbit Foot fire (Idaho, US) on 12 and 15 August

2018, and the Watson Creek fire (Oregon, US) on August 25, 2018. The flights were roughly 30 min to 1 h different from the TROPOMI overpass times. To take this into account, a plume age is estimated using the FLEXPART-WRF model, following the approach of Theys et al. (2020). The measurements are considered to be inside the plume if the $NO_2$ columns are greater than $3\times10^{15}$ molec/cm$^2$. Figure 10 shows the TROPOMI and aircraft comparisons: maps of both measurements are shown on the left panels with the VIIRS overlay, and the plume age of these measurements on the right panels for all three flights for

the TROPOMI $VCD_{EC}$ (original $VCD_{KNMI}$ can be found in the Appendix). There is good agreement between the aircraft and TROPOMI $VCD_{EC}$ $NO_2$ columns, the mean differences (CU-DOAS−TROPOMI) are -1.18±3.18×10$^{15}$ molec/cm$^2$ (-18 %) and -0.14±1.62×10$^{15}$ molec/cm$^2$ (7 %) (2.21±1.56×10$^{15}$ molec/cm$^2$) for the Rabbit Foot fire on August 12 and 15, respectively. The differences are calculated by estimating average aircraft columns that were observed within ±10 min of the TROPOMI plume age, as shown in Fig. 10 (right panels). There is not the best coverage of the Watson Creek fire for a good

comparison, as the aircraft measurements span a range between $3\times10^{15}$ and $3\times10^{16}$ molec/cm$^2$ and the time difference between the aircraft and the satellite is greater than 1 h.

From the CU-DOAS aircraft measurements $NO_2$ emission fluxes were estimated by integrating the columns for the entire plume transect and multiplying these by the wind speeds. Wind speed and direction were derived from in plume profiles made

in between plume underpasses. The results are summarized in Table 3 where the emission estimates for the EMG and the flux method are included using the $VCD_{EC}$ columns (the same table but using $VCD_{KNMI}$ is included in the Appendix). The emissions are lower when applying the flux method to the satellite observations, similar to the comparison with the FIREX-AQ emission estimates. In the sensitivity tests, however, the flux method derived emissions are biased high. This could be



due to the different lifetime of $NO_x$ in the model analysis compared to the real measurements, for very short lifetimes the flux method derived emissions can change significantly. When the same winds are used for the satellite- and aircraft-based emission estimates the agreement is very good between the two using the EMG method, but the emissions are underestimated with the flux method for the Rabbit Foot Fire on August 15 and the Watson Creek Fire. The Rabbit Foot fire measured on August

5    12, 2018 is the only fire where the TROPOMI emissions are high-biased compared to the aircraft emissions. However, this plume was measured further downwind (roughly 40 km) than the other fires and some of the $NO_2$ might have decayed, when accounting for the plume age (using a lifetime of 2 h), these emission estimates agree well with the emissions from the EMG method. The other two plumes were measured much closer to the fire (roughly 20 km). Some differences are also expected due to the different time of emissions; the CU-DOAS plume observed a plume age of roughly 2 h, and as discussed in the previous

10    section the time of emission for the TROPOMI estimates is 30±30 min prior to the overpass.





**Figure 10.** Comparison between the BB-FLUX aircraft measurements (CU-DOAS) and TROPOMI NO$_2$ VCDs (VCD$_{EC}$). The maps with the satellite pixels and aircraft transects are shown in the panels on the left (a, c, e). The overlay is a VIIRS true colour image with the MODIS fire hotspots, shown as red dots (obtained from NASA Worldview; https://worldview.earthdata.nasa.gov/). The plume age for pixels with VCD$> 1 \times 10^{15}$ molec/cm$^2$ is shown in the panels on the right (b, d, f) for the TROPOMI VCD$_{EC}$ (grey) and the CU-DOAS VCDs (yellow, orange and red). The time of the observations is displayed in the legend.





**Table 3.** Summary of the satellite (using $VCD_{EC}$ for the estimate) and CU-DOAS $NO_2$ emission estimates.

| Fire | TROPOMI EMG (t/h) | TROPOMI flux (t/h) | wind/lifetime corr. CU-DOAS (t/h) | wind corr. CU-DOAS (t/h) | CU-DOAS (t/h) |
|---|---|---|---|---|---|
| Rabbit Foot (12 Aug) | 3.5±1.3 | 1.4± 0.7 | 3.68±0.6 | 1.6±0.3 | 5.9±0.9 |
| Rabbit Foot (15 Aug) | 1.0±0.4 | 0.7± 0.4 | 1.9±0.8 | 0.7±0.2 | 1.8±0.4 |
| Watson Creek (25 Aug) | 2.2±0.8 | 1.2±0.6 | 6.4±1.7 | 2.8± 0.7 | 3.8± 1.0 |

## 6 Conclusions

Based on our analysis, we conclude that estimating biomass burning $NO_x$ emissions from single TROPOMI overpasses is possible with both a flux method and the EMG method, assuming that certain (low cloud cover, no pyrocumulus development, and consistent winds) conditions are met. Estimating biomass burning emissions from single overpasses is desirable as biomass burning emissions can change very quickly. Using synthetic data from an air quality model with prescribed emissions, we showed that the input emissions can be reproduced with either method. More consistent and better correlations are achieved with the EMG method, which also showed smaller uncertainties (38 %) compared to the flux method (53 %). The primary contributor to the uncertainties is the $NO_x$ lifetime, while winds contribute secondarily. It is important for wind speed and wind direction to be accurate, however, the EMG estimate is stable when the winds are a little inaccurate or uncertain. The largest uncertainties for the flux method result from: the method itself, the lifetime (only if the lifetime is short), and the uncertainty of the wind speed are the main contributors to the overall uncertainty. Using model output and aircraft observations, the $NO_2$ to $NO_x$ scaling that needs to be applied to (early afternoon) satellite-derived $NO_2$ emissions is stable for forest fires. Based on model results and aircraft measurements, TROPOMI-derived "emissions" of $NO_2$ should be scaled by a factor of 1.3 to 1.5 to obtain total emissions of $NO_x$ (which, at the point of emission, will largely be in the form of NO). For the $NO_x$ lifetime we derived $2 \pm 1$ h using the EMG for various fires. This is in good agreement with the results from the WE-CAN campaign that suggested a $NO_x$ decay time of 90 min in biomass burning plumes (Juncosa Calahorrano et al., 2021).

We further investigated the effects of an explicit aerosol correction on the AMF and consequently on the derived emissions. A comparison to aircraft-based integrated profiles and aircraft derived emissions showed improvement by using the aerosol-corrected AMFs over the original AMFs that rely on an implicit aerosol correction that assumes aerosols as clouds. Applying an explicit aerosol correction to the TROPOMI AMFs improves the TROPOMI $NO_2$ VCDs. The new $VCD_{EC}$ showed better agreement with aircraft observed VCDs over the standard product ($VCD_{KNMI}$). However, there is room for further improvement, as the version in this study approximates the AOD from the $NO_2$ enhancement. With the on-going development of a TROPOMI AOD product, this could be used in the future to improve the AMF.





There is agreement between the satellite-derived emissions using the flux method or EMG method and the aircraft derived emissions. The flux method always resulted in lower emissions compared to the EMG method, and usually underestimated the aircraft derived emissions during FIREX-AQ and the BB-FLUX campaign. There is better agreement when the EMG method is applied using the $VCD_{EC}$, and the aircraft- and satellite-derived emissions are typically within the estimated uncertainties.

We would recommend using the EMG method for estimating $NO_x$ fire emissions from TROPOMI single overpasses.

Overall, we conclude that fire emissions of $NO_x$ can be determined from the TROPOMI dataset and showing good agreement with aircraft-derived emissions. While this study focuses on forest fire emissions in North America, based on the availability of aircraft-borne measurements, fire emissions from TROPOMI can be derived globally and for different types of vegetation.

This can be helpful to evaluate the input emissions of air quality models and to determine an overall annual emission budget of wildfires. However, TROPOMI typically has a single daily overpass in the afternoon that can only provide limited information on the diurnal variability of the emissions. Future geo-stationary satellites, like the Tropospheric Emissions: Monitoring of Pollution (TEMPO) mission, will be able to give further insight into the diurnal variability, and the same methods can be applied to these observations. The combination of emission coefficients (amount of $NO_x$ per MW) together with the geostationary

GOES FRPs might also be useful to address the diurnal variability of fires and the total daily, monthly or annual emissions. As shown for the FIREX-AQ fires (Fig. 9), the GOES FRP is a good indicator of $NO_x$ fire emissions and tracks the emissions well. In a future study, we will look further into TROPOMI emissions and GOES-FRP to obtain more information on diurnal patterns and to obtain a total $NO_x$ budget from biomass burning in North America.

**Appendix A: Exponentially Modified Gaussian**

$$VCD_{NO2}(x,y,s) = a \cdot f(x,y) \cdot g(y,s) + B \tag{A1}$$

$$f(x,y) = \frac{1}{\sigma_1 \sqrt{2\pi}} \cdot exp(\frac{-x^2}{2\sigma_1^2}) \tag{A2}$$

$$g(y,s) = \frac{\lambda_1}{2} \cdot exp(\frac{\lambda_1(\lambda_1 \cdot \sigma^2 + 2y)}{2}) \cdot erfc(\frac{\lambda_1 \sigma^2 + y}{\sqrt{2\sigma}}) \tag{A3}$$

$$\sigma_1 = \begin{cases} \sqrt{\sigma^2 - 1.5y} & ,y < 0 \\ \sqrt{\sigma} & ,y > 0 \end{cases} \tag{A4}$$

$$\lambda_1 = \lambda/s \tag{A5}$$





$$erfc(x) = \frac{2}{\sqrt{\pi}} \int_{x}^{infty} \cdot exp(-t^2)dt \tag{A6}$$

$$t = -y/s \tag{A7}$$

5    The crosswind and downwind coordinates are described by $x$ and $y$ in km, the wind speed is in km/h, the plume spread (describing the width of the Gaussian plume) is $\sigma$ in km. $\lambda$ is the decay rate and the inverse of the lifetime $\tau$ $(= 1/\lambda)$ in h$^{-1}$. $B$ is the background column and $a$ is the enhancement factor in molec/cm$^2$. From $a$ the emission rate can be determined by $E = a \cdot \lambda$.

     The fit is performed in a python script using the SciPy package using the Levenberg–Marquardt algorithm, that minimizes
10   the difference between the fitted values and the observations.





## Appendix B: Aircraft comparison using VCD$_{KNMI}$

**Figure B1.** Same as Fig.9, but using the TROPOMI VCD$_{KNMI}$ instead for the emission estimate.







**Figure B2.** Same as Fig.10, but for the TROPOMI VCD$_{KNMI}$ instead. The mean difference is -1.75±4.51×10$^{15}$ molec/cm$^2$ (-26 %) and 2.21±1.56×10$^{15}$ molec/cm$^2$ (32 %) for the Rabbit Foot fire on August 12 and 15, respectively.



**Table B1.** Summary of the satellite (using $VCD_{KNMI}$ for the estimate) and CU-DOAS $NO_2$ emission estimates.

| Fire | TROPOMI EMG (t/h) | TROPOMI flux (t/h) | wind-adjusted CU-DOAS (t/h) | CU-DOAS (t/h) |
|---|---|---|---|---|
| Rabbit Foot (12 Aug) | 3.6±1.4 | 1.5± 0.8 | 1.58±0.3 | 5.88±0.9 |
| Rabbit Foot (15 Aug) | 0.6±0.2 | 0.7± 0.4 | 0.7±0.2 | 1.8±0.4 |
| Watson Creek (25 Aug) | 3.1±1.2 | 1.6±0.8 | 2.77± 0.7 | 3.8± 1.0 |

*Competing interests.* I declare that I or my co-authors have competing interests as follows: Rainer Volkamer is a member of the editorial board of AMT.

*Acknowledgements.* We would like to thank Andrew Weinheimer for his contributions to the WE-CAN measurements campaign. We acknowledge the Air Quality Research Division support teams and the National Research Council aircraft pilots and technical support team

5   for the aircraft measurement campaign. These measurements were carried out as part of the Oil Sands 2018 aircraft measurement campaign project, funded by the Oil Sands Monitoring (OSM) program by the Governments of Alberta and Canada. The authors would like to thank the FIREX-AQ, WE-CAN and BB-FLUX management team, aircraft pilots and crew of the aircraft. This work contains modified Copernicus Sentinel data. The Sentinel 5 Precursor TROPOMI Level 2 product is developed with funding from the Netherlands Space Office (NSO) and processed with funding from the European Space Agency (ESA). TROPOMI data can be downloaded from https://s5phub.copernicus.eu.

10  The MODIS data set was provided by LANCE FIRMS operated by NASA ESDIS with funding provided by NASA Headquarters. We would like to thank the many contributors to the development and generation of the ABI series on GOES-R, GOES-R data can be downloaded from https://www.avl.class.noaa.gov. CES, JP, IB, and PR acknowledge support in part by the NOAA Cooperative Agreement with CIRES, NA17OAR4320101.



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
