# Peer review of "Biomass burning nitrogen dioxide emissions derived from space with TROPOMI: methodology and validation"

_Atmospheric Measurement Techniques, 2021_

## Author Comment (AC1)

We would like to thank reviewer 1 for his/her comments, we have addressed these in the new version of the manuscript. We addressed each comment below and highlighted our answers in blue, the reviewer's comments are black.

**Reviewer 1:**

This paper investigated fire NOx emissions using TROPOMI NO2 observations. The authors also explored the impact of aerosol on TROPOMI NO2 observations and thus emissions by comparing against results derived from aircraft measurements. They concluded that a correction factor of 1.3 to 1.5 shall be applied to correct NOx emissions inferred from satellite NO2 observations. I would recommend some revisions before the publication.

General comments:

1. AMF calculation. Is there any specific reason to use both GEOS-Chem and GEM-MACH to calculate AMF? The authors mentioned that free tropospheric NO2 is not well represented in GEM-MACH. If so, does it make more sense to use GEOS-Chem for all layers? I'm worried that the usage of two models will introduce additional uncertainties.

   Only GEOS-Chem is used in the a priori to calculate the AMF, output from the GEM-MACH model is not used for the AMF estimate. Inside the plume, in the boundary layer a constant profile is used, based on the enhancement of the VCDs. Outside the plume simply the GOES-Chem profile is used representing background concentrations. Inside the plume the contribution from GEOS-Chem is actually very small.

   Our colleagues are currently developing the emissions for the free troposphere to input in GEM-MACH and hopefully in the near future we can use the upper tropospheric profile from just the GEM-MACH model. In this study, GEM-MACH is only used for the model sensitivity test of the two emission estimate methods to test if the methods can determine the true emissions (input emissions).

   We included the following in the manuscript (Sect. 3.1) to make this point clearer:

   "The amount in the free troposphere is on the order of $10^{14}$molec/cm$^2$ and small compared to the total tropospheric column inside fire plumes (~$10^{16}$ molec/cm$^2$). It is better to assume even a small amount of NO2 in the free troposphere when estimating AMFs than assuming 0."

2. Uncertainties of EMG method by assuming constant lamda and sigma. Please clarify the uncertainties in the manuscript.

   We added a little more details and discussion on the sensitivity of the EMG to the lifetime and plume spread in the appendix:

[Figure]

Figure A1. The impact of changing the lifetime and plume spread parameter on the slope of best fit (under a (i) scenario) using the EMG method to obtain the fitted emissions.

Using a different lifetime and plume spread does have an impact on the bias to the true emissions however the correlation is not affected by this. Note that changes in lifetime and plume spread can compensate each other. For the previous cases, discussed in Sect. 4.2 we use a plume spread and lifetime of 6 km and 1 h (note that this does not represent the true chemical lifetime).
Figure A1 shows the variation of the slope of best-fit of the fitted emissions to the true emissions: a lower lifetime will increase the emissions and a lower plume spread decrease the emissions. Thus, the emissions are almost identical when using σ = 9 km, τ= 2 h, and σ= 7 km, τ= 1.5 h. Based on this analysis, the uncertainty is about 25% within the associated spread of lifetimes and plume spreads. This is a major contributor of uncertainty, and thus it is important to find a realistic lifetime to reduce the overall uncertainties of the emissions estimate, which is not always easy.

3. "the difference of the lifetime between the model and the TROPOMI observations are expected, since the chemical lifetime of NO2 is shorter in the model compared to reality." I'm concerned about the robustness of this conclusion. It's likely that the winds in the model and reality differ significantly, which also causes the different plume shapes.

   Yes the winds could impact the plume spread and consequently the lifetime (as those go together – we have seen various examples using EMG for point sources that a high wind speed reduces the plume spread, but changes the lifetime to a lesser extend), however it is unlikely to impact a large sample of different locations and days. The winds between ERA5 (this not really measured, but the best we can get for a large variety of locations) and GEM can be different, but they do not have a systematic bias..

   There are a few reasons why the lifetime in GEM-MACH is shorter than in reality. However, this is not part of this study and needs further investigation as to what exactly is going on. Our colleagues are working on improving and developing the GEM-MACH model:

One possibility might be that the NOx cycling is shorter than it should be, since the number of organic reactions in the breakdown sequence is smaller than in reality. Another possible reason could be numerics and advection: if the source of emissions is less than 10 km in extent (the case for a 10km model), then there will be a tendency for the model's advection code to "smooth out" the plumes in the horizontal direction, at least until they grow a bit in size, not just due to the grid cells being much larger than the source size, but due to the inability of the model to resolve spatial gradients. The mass conservation algorithm will tend to "flatten" peaks in concentration.

Specific comments:

1. Page 2, line 31, I suggest reorganizing this paragraph, since the key message is not clear. I'm not sure whether the authors would like to emphasize the advantage or limitation of satellite observations.

   We have changed the paragraph to the following:

   "A few species can be observed by satellite instruments and used to estimate fire emissions. Satellite-remote sensing observations have the advantage of continuous, near global coverage, if meteorological conditions are favorable (e.g., clear-sky) and the emissions are above the instrument's detection limit.

   Ground-based and aircraft measurements are difficult to obtain near the fire source (due to Temporary Flight Restriction zones) and field campaigns are infrequent with limited spatial coverage, while satellite-borne observations can be used to constrain wildfire emissions and can provide emission estimates for fires missed by measurement campaigns."

2. Page 3, line 13. The 2011 work is based on OMI observations.

   Thank you for noticing this error. We have corrected the text accordingly.

"As satellites improved so did space-borne emission estimates, and in 2011 NOx emissions were derived for the first time on a city-wide scale (Beirle et al., 2011) using observations Ozone Monitoring Instrument (OMI; 2004–present; 13_24 km2; at nadir; Levelt et al., 2006; Krotkov et al., 2016)."

3. Page 3, line 19. Are there any differences between biomass burning investigated by Jin et al. (2021) and wildfire in this study? If not significantly, I would recommend a discussion or comparison with Jin's work in the manuscript since both studies use TROPOMI NO2 to infer NOx emissions. I notice the authors tries to do the comparison in the introduction by listing the topics covered by both studies. But I would appreciate some descriptions/clarification about differences, because it may be difficult for readers who are not familiar with Jin' work to understand the differences by just reading the list.

   As suggested, we have included further details in the introduction:

"Recently, TROPOMI-derived NOx emissions have been reported (Jin et al., 2021), which focused on TROPOMI-derived global NOx emissions and NOx emission factors. Our study explores the derivation of top-down NOx emissions from wildfires using TROPOMI NO2 observations and assesses its accuracies, with a focus on (1) the methods used for the emission estimates, (2) the conversion of retrieved NO2 to estimates of NOx, and (3) the explicit aerosol correction, and (4) validation of the TROPOMI-derived emissions using aircraft observations."

4. Page 4, line 5. please correct the typo of " the he".

   Done.

5. Page 4, line 30. What is the resolution after 6 Aug, 2019?

   3.5kmx5.5km. To make it clearer we changed the text to: "(3.5kmx5.5km after August 6, 2019; 3.5kmx7km prior to August 2019)"

6. Page 5, line 17. Is RPRO for the whole year of 2018? Please clarify here.

   We included the dates: "(RPRO; April to November 28, 2018)"

7. Page 5, line 27. What does the under script of EC stand for? Is it Environment Canada? It will give readers the impression that these are the official NO2 products from Environment Canada. Please consider renaming the products if it is not the case and the products are investigational. But this is just my feelings. Other readers may have different opinions about this. I would suggest ask around and make the final decision about the name.

   In previous studies the same terminology has been used. While this is not the official product of Environment Canada,  but it's unofficial product of Environment Canada. We decided to keep this name.

8. Page 5, line 33. please correct the typo of ". hourly".

   Done

9. Page 6, line 12. It is not clear to me how the model setup simplifies determining the accuracy of emissions estimation method.

   In the typical set up, the input fire emissions have a prescribed diurnal cycle. If the emissions change every hour it is harder to know the exact input emissions as these are variable and will impact the VCDs downwind of the fire (the VCDs are the result of ~3h of emissions, depending on wind speed and lifetime). If the input emissions are constant for the entire day, we know exactly the true emissions. We included the following in the text to make this point clearer:

"This removes the prescribed diurnal variability (used in the standard model run) and thus simplifies determining the accuracy of the emission estimation methods, as the input emissions are constant and known; concentrations downwind were emitted at the same rate as those close to the fire."

10. Is there any specific reason for only showing the flight track for AOSR, but not other three campaigns?

The flight tracks (at least for the flights used in this study) of the other three campaigns are shown in Figs. 9 and 10 together with the TROPOMI observations.

We have modified the text to point to those figures.

"Further details, including the flight path are presented in Sect. 5.3 (Fig. 10)."

"Further details, including the flight path are presented in Sect. 5.2 (Fig. 9)."

11. Figure 4. Please make the sizes of panels consistent.

The panels have the same size, however, the size on the left is determined by the ratio of the longitudes and latitudes.

---

## Author Comment (AC2)

We would like to thank reviewer 2 for his/her very detailed comments and interesting suggestions on how the AMFs can be improved. We addressed the comments in the new version of the manuscript. Details can be found below, our answers are highlighted in blue, the reviewer's comments are black.

We have included major changes to the AMF estimate, which now relies more on observations and less on assumptions. The changes include: (1) using the TROPOMI aerosol layer height as a proxy for the aerosol and NO2 profile within the plume, (2) a strong gradient on the aerosol and NO2 profile that seems more consistent with observations of profiles, (3) the AOD used for the AMF estimate is based on coincident VIIRS observations. The aerosol layer height is now included as a variable in the AMF lookup table.

**Reviewer 2 :**

The study uses TROPOMI observations to derive biomass burning NOx emissions. The authors apply two methods: flux and EMG, and suggest EMG is a better approach. The authors further evaluate satellite derived NOx emissions with aircraft measurements. Given the uncertainties of satellite retrievals, deriving NOx emissions from fire plumes is not an easy job. The authors have done a lot of work putting together satellite, models and aircraft measurements. I was expecting this study will represent a significant contribution to literature from the abstract, but I was a little disappointed after reading the whole manuscript.

Overall, I feel the authors made a lot of assumptions that are not justified or evaluated carefully. While the authors show their approach can somehow agree with measurements, I'm not really convinced whether these methods can be applied widely to other fires. Especially since this manuscript is under review at AMT, developing a solid, justified and widely applicable method is the key. I strongly recommend the authors carefully evaluate each of their assumptions. Since the authors put together a large dataset from aircraft campaigns. I think this paper will be useful if they can justify their assumptions with the aircraft measurements. Below are my detailed comments.

1. The novelty of this study is the explicit aerosol correction for AMF calculation, but I'm not convinced with the methods. There are many issues:

1) It's not clear to me why the authors only account for aerosol scattering effect. What about the aerosol absorption effect? The emissions of black carbon from fires should not be small. Aerosol scattering and absorption will influence AMF differently (Lin et al., 2015).

Yes, the aerosol absorption and scattering is accounted for in the SASKTRAN radiative transfer model. We corrected this in the text.

In the abstract: "…and effects of aerosol scattering and absorption on the satellite-retrieved…"

P.3, l. 29: "…is influenced by aerosol scattering and absorption…"

(2) It's not clear why the authors decide to use a constant profile shape for NO2 and aerosol. The authors simply assume constant NO2 and aerosol between surface and 2.5 km. Is this really a good assumption? Shouldn't the vertical profiles of NO2 and aerosol vary with meteorology, topography, fire injection height etc.? For aerosols, depending on the height of aerosol (above or below cloud), its impact on AMF should also be different.

We have changed our approach and included the TROPOMI AER_ALH (aerosol layer height) as an approximation of the aerosol and NO2 plume profile, as well as the AMF estimate. However, this is still will only be a simplification of the actual profile, which is necessary in order to apply this algorithm for a large number of fires (around the globe). It is computationally quite expensive to estimate a new AMF look up table for specific aerosol profiles, and will not be possible to be applied on a broader selection of fires.

We have now changed the profile of the aerosols to be constant until approximately the height of the TROPOMI Aerosol_layer, the same profile is used for the NO2 a priori profile (with a small contribution of GEOS-Chem (background) above the aerosol layer). This shape is a simplification, but broadly consistent with many fire profiles as shown in Griffin et al., 2020 (Atmos. Meas. Tech., 13, 1427–1445, 2020 https://doi.org/10.5194/amt-13-1427-2020, Fig. 1), and is computationally more efficient to estimate an AMF using a look up table. While model output could be used for this, it is highly uncertain (in terms of plume injection height, amount of emissions etc.), and also strongly depends on getting the location of the plume in the model right – we have previously investigated fire plumes from our model and often the plumes will not overlap exactly with the satellite observations, which would lead to a completely wrong profile shapes, e.g. background where there is a plume and vice versa.

The following changes were made in the manuscript (Sect. 3.1):

"To obtain a better understanding of the plume shape, we utilize the TROPOMI AER_LH product ("aerosol_mid_height"), and average these over the entire plume. Note, there is typically not good enough coverage to use the aerosol layer height for each TROPOMI NO2 pixel, thus, we use the average instead. Inside the plume we use a NO2 a priori profile that is well mixed between the surface and the TROPOMI aerosol layer height rounded up to the closest 500m (above ground), scaled by the standard KNMI VCDs (VCDKNMI):
$N(z) = n(z) \times (VCDKNMI - VCDabove)$; (3)
where $n(z)$ is the normalized profile shape, $N(z)$ is the new a priori NO2 profile used to estimated $nd(z)$ (in Eq. 4), and VCDabove is the VCD contribution above the plume. Between the aerosol layer height (rounded up) and 12 km, are background conditions and we use the concentrations from a monthly GEOS-Chem model run at the approximate time of the TROPOMI overpass on a 0.5_x0.67⁰ resolution version v8-03-01 (http://www.geos-chem.org, Bey et al., 2001; McLinden et al., 2014). We use the GEOS-Chem profile, as the free tropospheric NO2 is not well represented in GEM-MACH due to 30 missing elevated sources such as lightning and aircraft (Griffin et al., 2019). The NO2 amount above the plume is on the order

of 1014 molec/cm2 and small compared to the total tropospheric column inside fire plumes (~1016 molec/cm2). However, it is better to assume even a small amount of NO2 in the free troposphere when estimating AMFs than assuming 0. This a priori is a simplification of the true profile shape, however, (Griffin et al., 2020b) showed that the TROPOMI aerosol layer height will capture the main plume closer to the surface well, which is commonly well mixed. Prescribing a specific profiles based on observations is not practical to estimate a smoke plume specific AMF in an operational manner."

"The AMF(z) is the altitude-dependent AMF and is specific to each scene. Here, the SASKTRAN radiative transfer model (Bourassa et al., 2008; Zawada et al., 2015; Dueck et al., 2017) has been used to generate an altitude-dependent AMF look-uptable (LUT) for clear-sky (and cloudy conditions), as a function of solar zenith angle, viewing zenith angle, relative azimuthangle, surface pressure, surface albedo (cloud pressure), as well an AOD (for several values between 0 and 1), and top of the aerosol layer height. For simplicity, the aerosol profile is assumed to be well mixed between the surface and the TROPOMI aerosol layer height (rounded up) and is 0 above. This is a simplification of the aerosol profile shape, however, this shape is a good approximation of the bulk of the plume (if the plume is not elevated) (Griffin et al., 2020b) and is computationally cheaper for the LUT estimate."

(3) Equation 3 is also confusing. First, why use 3.5km as the cut off? Shouldn't this vary with boundary layer height? Second, VCD_KNMI uses NO2 profile from TM5 simulations, but VCD_freetrop uses NO2 profile from GEOS-Chem profile. How could the difference between VCD_KNMI and VCD_freetrop be used to scale the NO2 profile? Without further justifications, I feel such approach is arbitrary.

We have changed the approach and use the aerosol layer height as a proxy for the plume profile. Above the aerosol plume we use the GEOS-Chem monthly mean, which represents background conditions (which can be above or below the free troposphere).

To scale the NO2 profile shape, it doesn't really matter if the GEOS-Chem or TM5 is used (or even nothing at all), this is on the order $10^{14}$ and about 1 or even 2 orders of magnitude smaller than the enhancement. Also this is just an a priori, it does not need to be exact, it is by definition a best first guess.

(4) Does the monthly GEOS-Chem simulations include fire emissions? If not, what if the fire emissions are released into the free troposphere?

As mentioned above (point (2)), we have changed our approach and include the TROPOMI aerosol layer height as a proxy for the aerosol and NO2 profile shape. So if the plume went to higher altitudes this will be included and the profile will change accordingly. The GEOS-Chem profile above the plume (or aerosol layer height) reflects just average background conditions.

(5) The authors assume clear sky inside the plume, but outside the plume, they account for cloud conditions. I'm not convinced how good the assumption is here. I think this may lead to some inconsistency in the AMF or derived NO2 VCD within and outside the plume if the cloud and aerosols affect AMF differently. On the other hand, if clouds and aerosols affect

AMF in the same way within the plume, why do the authors use explicit aerosol correction rather than implicit correction?

The example is shown in Figure 2 f. It is challenging to quantify cloud fraction inside the plume and change it to an appropriate cloud fraction without the contribution of aerosols, we have not found a way of doing this. So while assuming no clouds (inside the smoke plume) might underestimate the impact of clouds if there are clouds in addition to the smoke aerosols. Assuming clouds (in addition to the aerosols) will definitely overestimate the effect of clouds in all cases. In the example shown there is no cloud (as seen by the VIIRS image Fig. 2 a), however due to the aerosols there is a cloud fraction, this should be 0 as there are no clouds. In many cases where clouds and smoke plumes overlap the cloud fraction will be larger than 0.5 and the observations will be filtered.

Outside the plume the conditions are different and thus the AMFs (and parameters that go into it) will also be different. Outside the plume: the profile shape is according to a background profile, and the aerosol contribution is close to 0 (AOD), but instead the cloud fraction is used to correct of clouds if there are any. If there are clouds outside the plume those will be clouds (as there are no or very little aerosols). As can be seen in Fig. 2d,e,f, outside the plume the VCDs from KNMI and our re-estimated VCDs are very similar – which gives confidence that the outside the plume AMF estimate is working well. The uncertainty of the cloud effect is included as part of the uncertainty of the AMF.

We changed the text accordingly to make this point clearer (Sect. 3.1):

"The cloud and clear-sky AMFs are only considered outside the plume, for the following reason: The cloud fraction has contributions from clouds and aerosols and cannot be entangled: Inside the plume, the aerosols are already accounted for, thus, if clouds are considered again, these smoke aerosols would be accounted for twice, explicitly and implicitly (as smoke is mistaken for clouds). So while assuming no clouds (inside the smoke plume) might underestimate the impact of clouds if there are clouds in addition to the smoke aerosols. Assuming clouds (in addition to the aerosols) will definitely overestimate the effect of clouds for all cases. Additionally, if clouds and smoke aerosols overlap the cloud fraction is more likely to be above 0.5 and will be consequently filtered. As such, inside the smoke plume (with the $VCD_{KMNI} > 1 \times 10^{15}$ molec/cm2) we assume clear-sky (cf= 0) and only correct for the smoke aerosols without the additional clouds. Considering the cloud fraction in addition to aerosols will lead to an increase in the NO2 VCD (Fig. 2f) that is considered as part of the AMF uncertainty. "

(6) The authors use NO2 VCD as a proxy for AOD, which is also confusing to me. They simply assume a constant relationship between NO2 and AOD, and the relationship is not at all evaluated in literature (Bousserez 2009 is not a peer-reviewed paper).

We agree, this is a valid point. We are no longer using this relationship, and have changed the approach. We now use the observed AOD from VIIRS at 445 nm instead. In the past we have investigated using MODIS, which has not a good enough coverage. We also received files of the TROPOMI NASA AOD product (TROPOMAER). While TROPOMAER (Torres et al., 2020) is in good agreement to VIIRS for cloud-free scenes, it is highly impacted by

clouds. Thus, we decided to use VIIRS AOD at 445nm (approximately the wavelength in which NO2 is retrieved, ~440nm).

We have made the following changes in the manuscript:

"We use the AOD retrieved from VIIRS on-board S-NPP at 445 nm (VAOOO) at 6 km resolution (Jackson et al., 2013), which is similar to the TROPOMI pixel size and the wavelength that NO2is derived (440 nm). The overpass time of S-NPP is similar to that of TROPOMI, within a few minutes. An example of the VIIRS AOD for the Williams Flats Fire on August 7, 2019 is shown in Fig. 2 b."

2. I'm not convinced that the assumption of constant lifetime and plume spread is valid. A recent study shows large variation of NOx lifetime in fire plumes (Jin et al., 2021). The spread of the fire plume should also vary with wind speed and fire intensity. Figure 3c clearly shows how the emissions would change with different assumptions of lifetime. While assuming constant lifetime is fine for the flux method, isn't the main idea of EMG method is to derive emissions and lifetime simultaneously while accounting for variation in plume spread (Lu et al., 2015)? If lifetime and spread is considered constant, EMG is essentially a smoothed exponential decay function, which is mathematically similar to the flux method. What's the motivation of using two methods then?

Using the EMG for single overpasses with variable sigma and plume spread reduces the amount of fires that can be retrieved by almost 70%. A sample of good fires is used to get a realistic lambda and sigma in this study (see Sect. 4.1). Note that sigma and lambda compensate each other. That is part of the reason for the sensitivity test. With this and a constant lifetime/plume spread for over 60 fires with various emissions and meteorological conditions, we can estimate the input emissions very well. The flux method does not depend on the plume width that is what changes the most for different wind speeds. Note that the EMG algorithm fails for large and spread out fires.

Tests show that retrieving a lifetime from fires using satellite observations is extremely impacted by the diurnal variability of fire emissions. E.g. the example of CO shows this very nicely. TROPOMI measures at 1:30 pm which is near the fire peak, in previous hours the emissions were lower as typically the fire builds up throughout the day. Here is an example from TROPOMI CO, according to that CO would have a lifetime of 3.0 h which is not real but solely due to the diurnal variability (and maybe dilution) and should not be used. For NOx there is this same diurnal pattern in addition to a short lifetime, so whatever lifetime is retrieved this is extremely influenced by the diurnal variability of the fire emissions. The estimate of the lifetime (bottom right panel) in this example figure is similar to the 1D EMG method as used in Jin et al., 2021.

[Figure]

Fig1: TROPOMI CO emissions example to show the effect of the diurnal variability on the lifetime. Note this is TROPOMI CO, no NO2 is shown.

We added the following to the discussion:

"When looking at fire emissions it is important to keep the diurnal variability in mind. TROPOMI measures at roughly 1:30 pm local time, at this time the fire activity is typically increasing (unless there is rain or the fire is extinguished), and emissions before the overpass were likely smaller than at the time of the overpass. This will impact the lifetime estimate using satellite observations and will likely not return the correct lifetime."

We have also included a section in the appendix that uses the EMG method freely (Appendix C: EMG without restrictions). Sometimes it works, but many times the method fails, and as mentioned the diurnal variability of the fire emissions has a huge impact on the lifetime estimate. Which is why we would recommend obtaining a good estimate of the lifetime first.

The two methods are commonly used and different, the EMG will use the plume shape to estimate the emissions taking the entire plume into account. Whereas the Flux method integrates over the VCDs to obtain an emitted mass, not considering the entire plume or plume shape.

1. The evaluation with aircraft measurements is new, but the comparison is overall limited to the statistics. For example, it's interesting to see statistically EMG performs better than flux methods, but why? Since the authors made the same assumptions with lifetime, I'm curious what factors could lead to such differences. Also, I guess the difference between TROPOMI and aircraft emissions is related to the short-term variability of fire emissions, which however is not discussed. These aircraft measurements may also help assess the assumptions made in AMF calculation, and I don't see any discussions on this.

Even though the lifetime is the same for the EMG method and the flux method, these are different methods and thus lead to different estimates of emissions, which are then compared to the aircraft-derived emissions. The EMG considers the shape of the plume, this is ignored with the flux method.

For the emissions derived from measurements during the FIREX campaign we limit the comparison when the time if emission was coincident. Included an additional sentence for the discussion on August 3, 2019 where the fire activity is changing rapidly:

"For the first TROPOMI orbit on 3 August, the emissions are very low, and for the second orbit, the fire activity then increased rapidly, which is likely why there are discrepancies between the satellite- and aircraft-derived emissions that day as the emissions changed very rapidly during this time."

For the comparison to the emissions from the BB-Flux campaign we are limiting the comparison to only include measurements that are close in time to the TROPOMI overpass to limit the impact of the diurnal variability of the fire emissions. However, the time of the emissions was greater than for the FIREX comparison. This was already discussed at the end of section 5.3.

We included the following in the conclusions:

"The diurnal variability also needs to be kept in mind when comparing to aircraft derived emissions, therefore, it is important to compare emission estimates for coincident times of emissions to limit the impact by the diurnal variability on the comparison. For the comparison between the TROPOMI-derived and aircraft derived emissions during FIREX-AQ and the BB-Flux campaign, we found agreement between the […]"

2. A lot of details are missing in terms of how the authors perform EMG. The authors simply listed a number of equations, but I'm not sure how to interpret these equations. What does each equation and parameter mean? How is implemented for each fire? I notice there is large data gap in Figure 4. How would this influence the EMG method?

The plume itself which is the most important part is well captured in Fig. 4 if there are not many observations around this doesn't impact the EMG significantly. A fit is performed to minimize the difference between the fitted and observed VCDs (this will ignore any locations without data and simply compare the VCDs that are actually observed).

The impact of missing data points is tested in the sensitivity test (Section 4.2) , where we use the model with perfect coverage (scenario (i)) and then also apply the satellite data gaps and perform the estimate; scenario (ii). From this the impact of the missing data points suggest that about 10% less firs can be retrieved due to satellite data gaps, and also leads to lower emissions compared to the perfect scenario with full coverage.

Also note that this is not the first paper to use this method, it has been published several times.

We added the following to Sect. 3.2.2:
"To describe the distribution of the NO2 VCD field near the source, an exponentially modified Gaussian(EMG) function can be used (see Eq. A1-4, in the Appendix). Using a Levenberg–Marquardt algorithm the enhancement factor a is derived by minimizing the difference between the fitted and observed VCDs. The enhancement factor is directly linked to the emissions E by:
$E = a/\tau$. (6)"

We have changed the description in the Appendix and hope this makes the EMG estimate clearer:

**Appendix A:  Exponentially Modified Gaussian**

The EMG method describes a Gaussian shaped plume in cross-wind (x) and along-wind (y) direction. The fit is performed in a python script using the SciPy package using the Levenberg–Marquardt algorithm, that minimizes the difference between the fitted VCDs and the satellite observed VCDs, where we use Eq. A1 and find the best solution (for a, B; and occasionally $\lambda$ and $\sigma$- depending if these are held constant or are fitted, as described in the text) with scipy.optimize.curve_fit (method="lm"). The following equations are used to describe the Gaussian plume, the wind speed is needed for this, the decay rate $\lambda$ (inverse of the lifetime) can either be fitted or can be a fixed parameter, similarly the plume spread $\sigma$ can be fitted or a fixed parameter.

$$VCD_{NO2}(x,y,s) = a \cdot f(x,y) \cdot g(y,s) + B \quad (A1)$$

From the enhancement factor a, the emissions can be determined by $E = a \cdot \lambda$. The functions $f(x,y)$ and $g(y,s)$ describe the plume shape:

$$f(x,y) = \frac{1}{\sigma_1 \sqrt{2\pi}} \cdot \exp\left(-\frac{x^2}{2\sigma_1^2}\right) \quad (A2)$$

$$g(y,s) = \frac{\lambda_1}{2} \cdot \exp\left(\frac{\lambda_1(\lambda_1 \cdot \sigma^2 + 2y)}{2}\right) \cdot erfc\left(\frac{\lambda_1 \sigma^2 + y}{\sqrt{2}\sigma}\right) \quad (A3)$$

The crosswind and downwind coordinates are described by x and y in km, the wind speed,s is in km/h, the plume spread(describing the width of the Gaussian plume) is $\sigma$ in km. $\lambda$ is the decay rate and the inverse of the lifetime$\tau (= 1/\lambda)$ in h$^{-1}$, and $\lambda_1$ is short for $\lambda/s$ (inverse of the lifetime over the wind speed). B is the background column (fitted) and a is the enhancement factor in molec/cm$^2$. erfc is complementary error function and is included in scipy (scipy.special.erfc). $\sigma_1$ is described differently upwind and downwind of the fire hotspot:

$$\sigma_1 = \sqrt{\sigma^2 - 1.5y} \ , y < 0 \quad \sqrt{\sigma}, y > 0 \quad (A4)$$

Further details about the EMG can also be found in other publications, e.g. Fioletov et al. (2015) and Dammers et al. (2019).

Specific comments:

Page 3 Line 24: You already mentioned TROPOMI in previous paragraph.

Changed the text recommended.

Page 12 Line 30: Here you mentioned using TROPOMI aerosol layer height for wind speed, but why not use this information in aerosol correction for AMF?

As described above this has been changed as recommended, we now incorporated the TROPOMI ALH in the AMF look up table.

Page 13 Figure 2: Here VCD_EC seems to be much smaller than VCD_KNMI, but Figure 8 shows the opposite. I understand that aerosol may influence AMF differently. To avoid confusion, I'd suggest the authors either limit to one fire case (the case with aircraft), or explain under which conditions aerosol corrections lead to higher VCD and vice versa.

As suggested, we included a different example where the re-calculated VCD increases. The relationship is not simple and strongly depends on the viewing geometry etc. We included a figure in the appendix showing the AMF for two different scenarios.

"An example of the NO2 tropospheric VCDs, with and without an explicit aerosol correction, is shown in Fig. 2 for the Williams Flats Fire on August 7, 2019. Figure 2 (a) displays the VIIRS true colour image at approximately the same time as the TROPOMI overpass together with the MODIS thermal anomalies (red dots), showing no clouds over the fire plume. Figure 2 (b) shows the VIIRS AOD. The cloud fraction can be seen in panel (c), showing that the smoke plume is identified as clouds. The NO2 VCD KNMI, and VCD_EC are shown in panels (d), and (e), respectively. This illustrates that the NO2 VCD can change significantly when the AOD is accounted for, in this example, the NO2 VCDs increase over the fire hotspot. Note that the explicit aerosol correction can increase or decrease the VCDs, the relationship is not a simple linear relationship, it depends on the viewing geometry and AOD (see Fig B1 in Appendix B for more details). In this example, accounting for clouds in addition to the smoke aerosols is probably incorrect, as there were no clouds mixed with smoke. Figure 2 (f) shows the VCDs if both the aerosols and cloud fraction are considered in the estimate: this increases the NO2VCD (in this case, again this can go either way depending on the viewing geometry and AOD) in comparison to assuming no clouds, as in Fig. 2 (e). Panel (f) is only shown for comparison purposes, this approach accounts for smoke aerosols twice, explicitly and implicitly, and is therefore not recommended. Outside fire plumes, where the NO2is at background levels, as expected, the VCD_KNMI and VCD_EC are very similar."

[Figure]

Figure B1.This figure illustrates the changes in AMF with changing AOD. The altitude dependant AMF profile is shown in panel a and c for an aerosol layer height of 3 km with a changing AOD. Panel b and d show the total integrated AMF versus the AOD using a typical temperature profile. The difference between the top panel and bottom panel is the viewing geometry. Top panel: VZA=50◦, SZA=50◦, and DAZ=60◦and bottom panel: VZA=70◦, SZA=78◦, and DAZ=150◦. The surface pressure is 1000 hPa and the albedo is 0.09 in both examples.

Page 14 Line 20: Why did you choose 20 km for box size? It seems that the fire plume goes much further than 20 km in Figure 3?

Yes it does go further but moving further along doesn't help it the estimate of the fire emissions. As the mass downwind is accounted for, the emissions will vary due to the

diurnal variability of the fire emissions. 20 km is chosen to ensure the entire fire is considered and enough TROPOMI observations are averaged, with emissions that have been emitted within roughly 1 h of the satellite overpass.

We included the following in the manuscript to make the choice of the 20 km clearer:

"To obtain the final number for the emissions only the boxes within 20 km of the fire are averaged (Adams et al., 2019), which ensures that the entire fire is captured and enough TROPOMI observations are used for the estimate. Limiting the estimate to within 20 km of the fire reduces the impact of the diurnal variability of the fire emissions, on average the NOx molecules within 20 km were emitted roughly within 1 h of the TROPOMI overpass."

Page 17 Line 20: Did you look same fires for TROPOMI and GEM-MACH? Why NO2 lifetime is shorter in model than observations? Maybe it's due to different resolutions? Also, what's the chemical lifetime of NO2 in GEM-MACH?

This is not an issue due to the resolution, we obtain similar lifetimes for TROPOMI and OMI (which has a broader resolution than the GEM-MACH model) estimates.

There is no prescribed lifetime of NOx in the model itself, it depends on the chemistry and the parametrization within the model.

There are a few reasons why the lifetime in GEM-MACH is shorter than in reality. However, this is not part of this study and needs further investigation as to what exactly is going on. Our colleagues are working on improving and developing the GEM-MACH model:

One possibility might be that the NOx cycling is shorter than it should be, since the number of organic reactions in the breakdown sequence is smaller than in. Another possible reason could be numerics and advection: if the source of emissions is less than 10 km in extent (the case for a 10km model), then there will be a tendency for the model's advection code to "smooth out" the plumes in the horizontal direction, at least until they grow a bit in size, not just due to the grid cells being much larger than the source size, but due to the inability of the model to resolve spatial gradients. The mass conservation algorithm will tend to "flatten" peaks in concentration.

Page 18 Line 33: The errors of satellite retrievals are not necessarily random. Studies have reported low bias of TROPOMI NO2.

The systematic error will affect the emission estimate, and a low bias of the VCDs will of course decrease the emission estimate, thus this has not been tested as part of the sensitivity tests as this outcome is predictable.

We estimate a new AMF as part of this study to correct for the low bias found in TROPOMI.

We included the following sentence in the manuscript:

"On the other hand, if there is a bias in the satellite VCDs those will affect the emission estimate, this can however be corrected with re-calculated AMFs and high-resolution input data."

Page 23 Line 2: Do you assume constant lifetime and spread here? If so, why not try relaxing these assumptions for synthetic observations, and see wether original EMG method still works?

We have included a section in the appendix (Appendix C: EMG without restrictions) showing the emission estimate using the EMG that estimates the lifetime and plume spread. As can be seen in the sensitivity test, while for some fires the true emissions can be estimated, wrong emissions are estimated for many other fires. We have also included two tables with the emission, lifetime and plume spread estimates for the fires during BB-FLUX and FIREX-AQ campaign.

Table 1: I think there are other sources of uncertainties not discussed here. Just to name a few: 1) uncertainties of your AMF method (especially with prior); 2) uncertainties of the aerosol information; 3) biases in satellite retrieval of NO2 columns; 4) uncertainties in the plume injection height.

These uncertainties are included: Uncertainties of AMF included ~20% (which includes AOD, aerosol layer height etc.), also the bias in the satellite "retrieval" is part of the AMF uncertainty. The bias of the satellite retrieval is really the bias of the AMF, if the AMF is correct the VCD will not be biased.

The uncertainty of the layer height is part of the uncertainty of the winds, bias of the NO2 part of the AMF. To be honest, I think 40% is already an overestimate of the uncertainties as many tend to compensate each other, and considering that the aircraft and satellite emissions estimates are in such good agreement.

We included the following to make this clearer in the manuscript:

"The uncertainty of the wind speed also includes the uncertainty of the wind heights used to obtain the wind speed. […] The uncertainty of the satellite VCDs is really the uncertainty of the AMF which includes the uncertainties related to the assumptions and parameters used for the calculation of the AMF."

Figure 8c: It looks like there is large gradient of NO2 at low altitude, which differs from the interpolated profile. This again made me doubt about the validity of your assumption with the NO2 profile. Also, it's better to present vertical profiles in pressure gradient, which could better show the vertical gradient of NO2 at lower altitude.

As mentioned above we have changed the profile and now se a profile with a steep gradient that better represents profiles near fires.

Page 26 Line 11: While NO2 columns downwind are less important for overall enhancement, this would impact on the lifetime of NO2.

Yes it does, but here we use a constant that has been obtained before.

Page 27 Line 3: Did you account for diffusion when calculating NOx emissions from aircraft measurements?

If by diffusion the physical spread/distribution of NOx is referred to then this method takes it into account. The spatial distribution of emissions is accounted for since it utilizes the smoke spatial distribution retrieved by lidar extinction measurements. This might be more targeted at in situ measurements that only capture emissions at one point in the smoke plume and as we've seen the concentration of compounds can vary depending on whether you are sampling the center versus the edge, etc. This method avoids some of that sampling bias.

Page 29 Line 7: Please provide justification for the threshold.

This is an arbitrary threshold simply picked so that only VCDs inside the plume are compared that are not too close to the plume edge with strong VCD gradients.

We included the following in the manuscript:

"The measurements are considered to be inside the plume if the NO2 columns are greater than $3x10^{15}$ molec/cm$^2$, this threshold has been chosen to avoid measurements too close to the plume edge."

Figure 10: The plot looks messy. Why not just show the mean and standard deviation of NO2 column from CU-DOAS?

We have changed the plot and only show the 10s averages (together with the standard deviation) for the CU-DOAS measurements.

Figure 10: I suggest include TROPOMI VCD_KNMI here.

As suggested, we included the estimates using the VCD_KNMI in the figure and removed the corresponding figure from the Appendix.

Page 29 Line 18: Not clear what you mean here. Emissions = columns x wind speed? It doesn't sound right to me.

It is not column x wind speed, as described in the manuscript, the columns are integrated across the plume (see text) times the wind speed. See for example Theys et al., 2020 (https://www.nature.com/articles/s41561-020-0637-7) or Varon et al., 2018 (Atmos. Meas. Tech., 11, 5673–5686, 2018, https://doi.org/10.5194/amt-11-5673-2018).

Page 32 Line 13: It's not clear where the scale factor of 1.3 to 1.5 comes from. Table 2 shows large difference between TROPOMI EMG and aircraft derived emissions, and the

difference also varies a lot fire to fire. I don't think it's correct to suggest a universal scale factor.

We included the following in Sect. 4.3 to make it clearer where the 1.3 -1.5 comes from:

"For studies looking to convert satellite derived NO2 emissions to NOx emissions, based on this analysis, we recommend using a value between 0.68-0.75 for day time satellite derived NO2 emissions (for early afternoon overpasses), and thus, applying a factor of 1.3 (=1/0.75) to 1.5 (=1/0.68) to the satellite-derived NO2 emissions to obtain the net NOx emissions."

Page  32 Line 15: I'm confused here. Didn't you assume constant lifetime for EMG approach (Page 15 Line 5)?

Yes that is correct, but we also estimated the lifetime and plume spread for a number of "good" fires, see Sect. 4.1, p. 18 l. 20-25.